# A Comprehensive Study of the Use of LoRa in the Development of Smart Cities

**Roberto Omar Andrade** [1,2] **and Sang Guun Yoo** [1,2,*] 

[1] Facultad de Ingeniería de Sistemas, Escuela Politécnica Nacional, Quito 170525, Ecuador; roberto.andrade@epn.edu.ec

[2] Smart Lab, Escuela Politécnica Nacional, Quito 170525, Ecuador

[*] Correspondence: sang.yoo@epn.edu.ec

**Abstract:** The New Urban Agenda (Agenda 2030) adopted at the United Nations Conference related to Sustainable Urban Development (Habitat III) in the year 2016 has the goal of prompting cities to achieve the identified Sustainable Development Goals by the year 2030. In this context, cities can experiment strategies of circular economy for the optimization of resources, waste reduction, reuse, and recycling. The data generated by the components of an Internet of Things (IoT) ecosystem can contribute in two relevant ways to a smart city model: (1) by the generation of a circular economy and (2) by the creation of intelligence to improve the decision-making processes by citizens or city managers. In this context, it is in our interest to understand the most relevant axes of the research related to IoT, particularly those based on the LoRa technology. LoRa has attracted the interest of researchers because it is an open standard and contributes to the development of sustainable smart cities, since they are linked to the concepts of a circular economy. Additionally, the intention of this work is to identify the technological or practical barriers that hamper the development of solutions, find possible future trends that could exist in the context of smart cities and IoT, and understand how they could be exploited by the industry and academy.

**Keywords:** IoT; LoRa; LPWAN; smart city; circular economy

## 1. Introduction

The New Urban Agenda (Agenda 2030) adopted at the United Nations Conference related to Sustainable Urban Development (Habitat III) in the year 2016, has the goal of prompting cities to archive the identified Sustainable Development Goals by the year 2030. The Agenda 2030 states that "well-planned and well-managed urbanization can be a powerful tool for sustainable development". In this context, cities can experiment strategies of circular economy for the optimization of resources, waste reduction, reuse, and recycling [1].

Under the perspective of a circular economy, a city focuses on waste management, keeps assets, and enables digital technology. A study [2] identified several sectors for the implementation of a circular economy in a city. Those sectors are: built environment, energy, mobility, waste, toilets, industrial production, agri-food, citizens, and communities. Indicators and criteria of circularity in built environments proposed in this study [2] are shown in Table 1.

The inclusion of IoT is considered an important enabler of circular economy in a smart city. The above-mentioned study [2] considers IoT relevant to producers and designers, as they can obtain feedback in real time about products and infrastructures and keep materials in constant circulation. On the other hand, the work about IoT discussed by the Ellen MacArthur Foundation in the World Economic Forum mentioned that intelligent assets can sense, communicate, and store information by

themselves. Therefore, this could lead to products that can monitor their deterioration and alert the user, figure when they need to be repaired, and schedule their own maintenance [3].

**Table 1.** Criteria and indicators of circularity in built environments [2].

| Criteria of Circularity | Indicator of Circularity |
|---|---|
| Reuse of old buildings | Number of existing reused buildings |
| Energy efficiency | % of energy reduction in buildings |
| Greenhouse gas emissions avoided | % $CO_2$ emissions avoided (tons/year) |
| Water consumption avoided | % of water use reduction in buildings |
| Waste avoided in the construction | % of waste reused |

The smart city intends to convert its existing infrastructure into intelligent goods that are capable of providing enough information to make decisions more efficiently in real time. To achieve this goal, millions of sensors around the globe need an efficient, long-range, and low-energy-consuming technology. Low-Power Wide-Area Network (LPWAN) solutions like LoRa, SigFox, and NB-IoT meet these requirements [4].

The data generated by the components of an IoT ecosystem can contribute in two relevant ways to a smart city model: (1) in the generation of a circular economy and (2) in the creation of intelligence to improve decision-making processes by citizens or city managers.

The aim of this study is the comprehension of technological and functional aspects of IoT based on LoRa, which could support the development of a circular economy and help to generate smarter solutions for decision-making. LoRa has attracted the interest of researchers because it is an open standard, which contributes to the development of sustainable smart cities, since they are linked to the concepts of circular economy. Additionally, the intention of this work is to identify the technological or practical barriers that have limited the development of solutions and to find possible future trends that could exist in the context of smart cities and IoT and how they could be exploited by the industry and academy.

The rest of the present paper is structured as follows. Section 2 presents the state of the art of smart city, IoT, and LoRa technology. Section 3 presents the method used for the qualitative analysis. Section 4 presents an analysis of the results obtained from a systematic literature review (SLR), with the purpose of determining the main aspects that have made LoRa one of the most used technologies in the development of smart cities applications. Then, Section 5 discusses the strengths and weaknesses of LoRa-based implementations, and finally Section 6 presents our conclusions.

## 2. Background

### 2.1. Dimensions and Projections of Smart Cities

Urbanization brings several advantages to the society but also implies big challenges for city managers, since it increases the demand for energy, water, sanitation, education, health services, and other public services. Statistical data indicate that 4.50 billion of the world population, which is close to 68% of the global population, will live in cities by 2050 [5], and 426 million people will be over 65 years of age by the same year [6]. Additionally, projections indicate that 80% of the global energy will be consumed in cities in 2030 [7], and about $7 trillion will be invested by 2050 in the automobile sector, especially in self-driving cars [8]. In this situation, the adoption of smart cities models to make cities more sustainable, competitive, socially cohesive, and safe has become an indispensable task [9].

Smart city models are based on the idea of sustainable development considering people as the central axis. They has six pillars: governance, economy, living, environment, people, and mobility [10] (see Figure 1). From this perspective, city managers' decision-making processes can be improved through the sensorization of the different aspects of the city such as environment, safety, traffic, pollution, and so on, for mitigating and preventing urban challenges. For example, Figure 2 shows a

smart city sensor model, which considers the sensorization of different aspects related to industrial automation, pollution, energy, and traffic; the model also includes social interactions generated by people through different social network channels and components of data analytics using technologies such as cloud computing, big data, and high-performance computing (HCP).

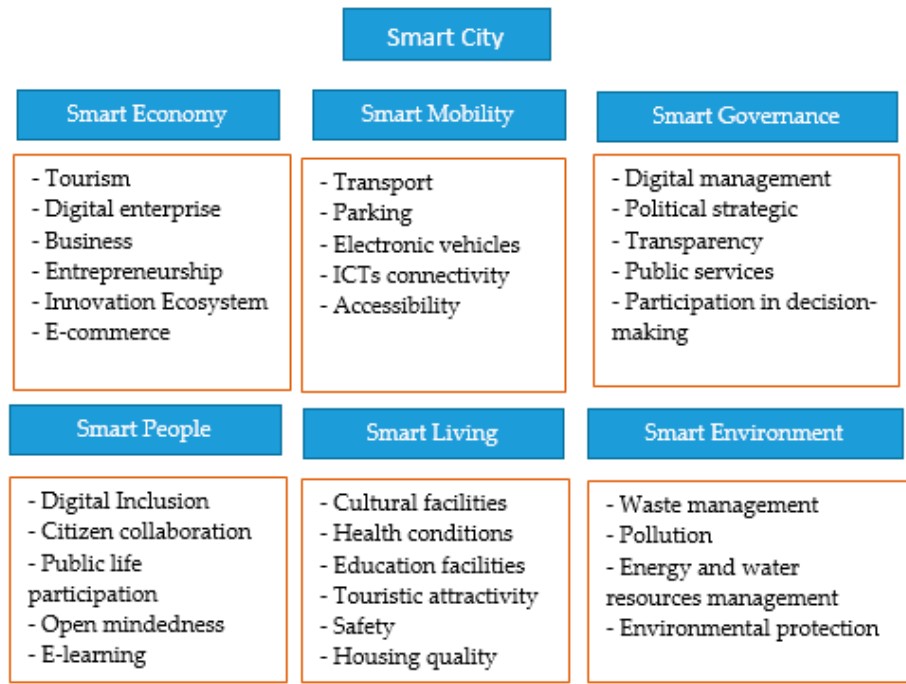

**Figure 1.** Dimensions in the development of an effective smart city model [10].

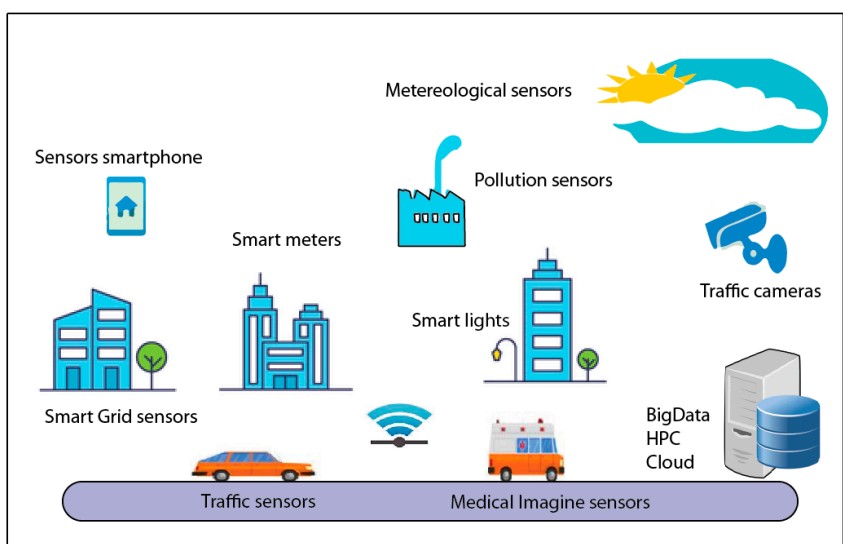

**Figure 2.** Sensorization and communication technologies (ICT) integration for the development of smart cities.

In order to establish the sensorization of a city, smart cities require information and communication technologies (ICTs) [11], specifically, an emergent technology called Internet of Things (IoT) that allows a deeper level of sensorization. According to a study published by Gartner in 2015, the number of devices connected was close to 1.1 billion in 2015 and predicted to be approximately 2.6 billion in 2017 [12]. The same study indicated the biggest subcategories as the smart home, with approximately 1.067 million connected devices, and the smart building, with 648.1 million devices (see also Table 2).

**Table 2.** Connected devices in the smart city subcategory (in millions).

| Smart City Subcategory | 2015 | 2016 | 2017 |
|---|---|---|---|
| Healthcare | 9.7 | 15 | 23.4 |
| Public Services | 97.8 | 126.4 | 159.5 |
| Smart buildings | 206.2 | 354.6 | 648.1 |
| Smart homes | 294.2 | 586.1 | 1.067 |
| Transport | 237.2 | 298.9 | 371 |

Source: Gartner 2015.

On the other hand, a report published by STATISTA in 2019 indicated that the number of IoT devices in the smart city sub-category was 260.6 million in 2015, 314 million in 2016, 380.6 million in 2017, and 463.5 million in 2018 [13]. These numbers re lower than those estimated by Gartner in 2015. However, they still reflect the significant growth of IoT implementations for smart cities. Gartner, in another study published in 2018, estimated that IoT would grow to reach 1.1 billion devices in 2019 and close to 9.7 billion by 2020. This study also indicated that IoT will expand the usage of new technologies such as smart machines and robotics [14]. According to a McKinsey's study, IoT technologies will have a potential economic impact of $2.7 to $6.2 trillion by 2025 [15].

### 2.2. IoT Technologies for Smart Cities

It is known that IoT applications for smart cities require an adequate network connectivity with large area coverage and efficient resource consumption, especially in terms of energy, since most IoT devices use battery for their operation. The low-power wide-area network (LPWAN) was created to satisfy these requirements. LPWAN is a group of technologies that allow long range communications with low power consumption, long battery life (approximately 10 years), acceptable transmission rates, good penetration characteristics, and reduced cost of installation [12]. There are different technologies that represents LPWAN (see Figure 3). Among these, LTE-M, EC-GSM, NB-IoT are considered as 3rd Generation Partnership Project (3GPP) standards, whereas SIGFOX, LoRa, and Weightless are classified as non-3GPP standards. NB-IoT (narrow-band IoT) was released in June 2016 and was developed by the 3GPP consortium; it is compatible with LTE and has a coverage range of 15 km and a transmission speed of around 150 kbps. NB-IoT is a licensed technology which has been deployed in 79 countries with an investment of $598 billion. NB-IoT uses OFDMA modulation. On the other hand, LoRa was developed by Cycleo and was released in 2012 as a non-licensed technology. It has a range of 11 km and a transmission speed of around of 10 kbps. LoRa networks have been implemented in 34 countries with an investment value of $241 billion. Finally, Sigfox is a non-licensed technology which was developed in Toulouse and released in 2009. It has a coverage range of 13 km and a transmission speed of 100 bps [16].

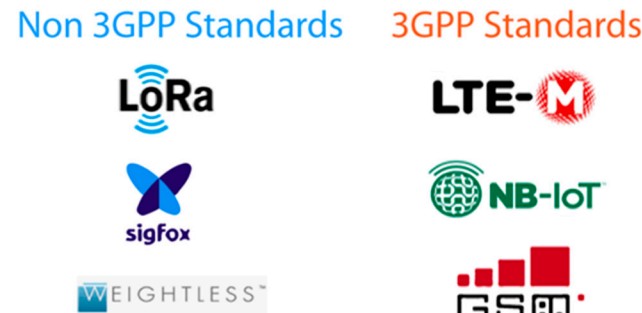

**Figure 3.** Classification of Internet-of-Things (IoT) communication technologies from the 3rd Generation Partnership Project (3GPP) perspective [17].

Among the different LPWAN technologies, LoRa was considered for this study since it operates efficiently in unlicensed bands. Other technologies have various limitations in terms of research. For example NB-IoT must be setup within a cellular network [18], Sigfox is a non-licensed technology that operates on the basis of subscription; therefore, to connect devices to the Sigfox network, the usage of application programming interfaces (APIs) of the Sigfox cloud platform is required. Other advantages of LoRa are that it is supported by many worldwide technology leaders, it has two model services, i.e., a provider network service and a private network, and the number of messages that can be sent in the LPWAN infrastructure is not limited (whereas Sigfox only allows 140 messages per day and per device) [19].

Other important characteristics of LoRa are: (1) LoRa presents robustness against multi-path fading or Doppler effect; (2) LoRa also uses a proprietary chirp spread spectrum (CSS) modulation to archive distances greater than 700 km, and the CSS modulation provides robustness against interference and a very low signal-to-noise ratio (SNR) that allows the receiver to demodulate the signal; (3) LoRa is suitable for applications that require a very long battery lifetime and has a reduced installation cost; (4) LoRa allows tuning several physical transmission properties, such as bandwidth (BW), central frequency, coding rate (CR), spreading factor (SF), and transmission power [18]; (5) LoRa uses a long-range start architecture (see Figure 4) in which gateways are used to relay messages between end nodes; (6) LoRa uses an adaptive data rate (ADR) algorithm to estimate the CR and SF parameters under a specific channel.

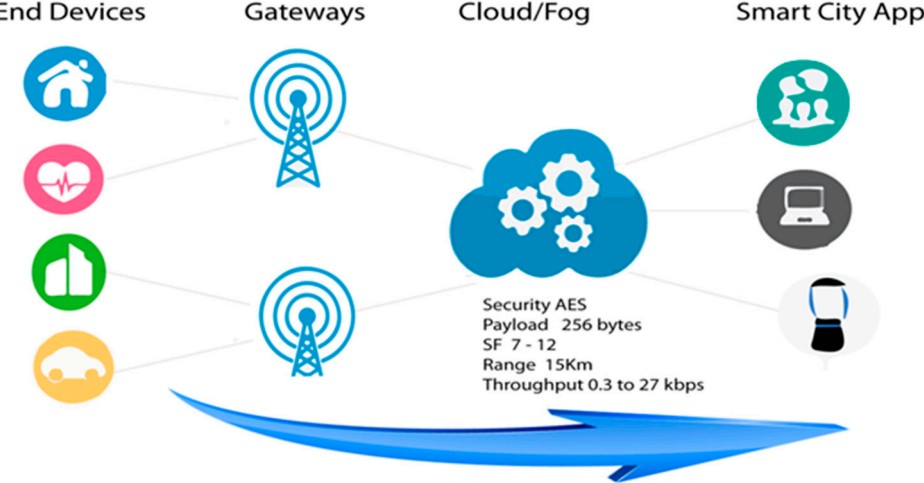

**Figure 4.** LoRa's generic architecture: end nodes, gateways, cloud/fog, and apps.

### 2.3. Parameters of Transmission for LoRa

The implementation of LoRa must be aligned with the functional and operational requirements of each subcategory of the smart city, which implies considering aspects such as real time, security, traceability, integration with GPS, 3D, and drones solutions. This entails considering technical aspects such as frequency of operation, bandwidth, or throughput, but it also implies taking into account elements inherent to the environment, such as the existence of buildings, trees, or other obstacles that affect the line of sight. A literature review indicated that LoRa modulation has a direct relationship with DR (data rate), SF, and BW. The selection of these parameters must be aligned with the functional and technical requirements of IoT solutions. DR is a relationship between SF and BW and can be expressed as follows:

$$DR = SF * \frac{1}{\frac{[2SF]}{BW}} \text{ bits/ sec}$$

The transmission time or time on air is the sum of the preamble transmission time and the physical layer's message transmission time, which depends on the symbol time that is directly associated with the spread factor and bandwidth.

$$Ttx = Tpreamble + TPHYmessage$$

$$Tpreamble = Ts * (Npre + 4.25)$$

where *Npre* is the number of symbols to be used in radio transceiver; the symbol period *Ts* is expressed as follows:

$$Ts = \frac{2SF}{BW}$$

$$TPHYmessage = TS * NPHY$$

*NPHY* is the number of symbols transmitted in the physical layer and can be determined as:

$$NPHY = 8 + \max\left[ceil\left[\frac{28 + 8PL + 16 * CRC - 4SF}{4 * (SF - 2DE)}\right] * (CR + 4), 0\right]$$

where *CR* denotes the coding rate which can take values from 1 to 4, and *PL* is the payload rate that indicates the physical payload length in bytes, which is related to the type of application in a smart city. Cyclic redundancy check (CRC) indicates the presence of an error detection field, and DE indicates that the low data rate is enabled.

The signal values received in IoT solutions must be appropriate for the sensitivity levels of the transceivers. The IoT ecosystem in LoRa networks should consider aspects related to losses and gains from the transmitter to the target receiver:

$$Prx(dBm) = Ptx(dBm) + Gsystem(dB) - Lsystem(dB) - LChannel(dB) - M(dB)$$

where *Gsystem* and *Lsystem* correspond to the gains and losses associated with antennas and cables. *LChannel* is associated with losses related to the transmission medium that will also be dependent on environmental conditions or reliefs in the geographical area. *M* is related to the fading margin. Also, in indoor implementations, noise floor should be considered:

$$Noise\ Floor = 10 * log10(k * T * BW * 1000)dBm$$

where *K* is Boltzmann's Constant, and *T* is 293 Kelvin or room temperature.

Summarizing, the most relevant characteristics of LoRa are:

- High sensitivity in receiving data (End nodes: Up to −137 dBm, Gateways: Up to −142 dBm)
- High tolerance to interferences: resistant to the Doppler effect, multi-path fading, and signal weakening
- Strong indoor penetration. High SF, up to 20 dB penetration
- Low consumption of energy, approximately 10 years of battery lifetime
- Low range of coverage, between 10 to 20 km
- Reduced data transfer
- Point-to-point connection and Gateway connection
- High Scalability. Only one gateway can cover a radius of 15 km
- Two models of service: network provider or private network

As indicated before, LoRa is a robust non-licensed open technology, and these advantages have motivated researchers to use LoRa in their projects covering urban and rural areas solutions. The aim of the present study is to carry out a comprehensive study of LoRa's impact in developing IoT solutions for different categories of the smart city.

## 3. Materials and Methods

The development of IoT and smart cities has been considered by international organizations, such as the International Union of Telecommunications (ITU), the Institute of Electrical and Electronics Engineers (IEEE), the International Standardization Organization (ISO), and byconsulting firms, such as Gartner and Forbes. Among them, ITU in its recommendation Y.2060 has proposed an IoT architecture that supports smart city subcategories such as smart transport, smart energy, smart health, and smart living (see Figure 5). The architecture considers aspects like IoT devices, network capabilities, services support, and security issues [20].

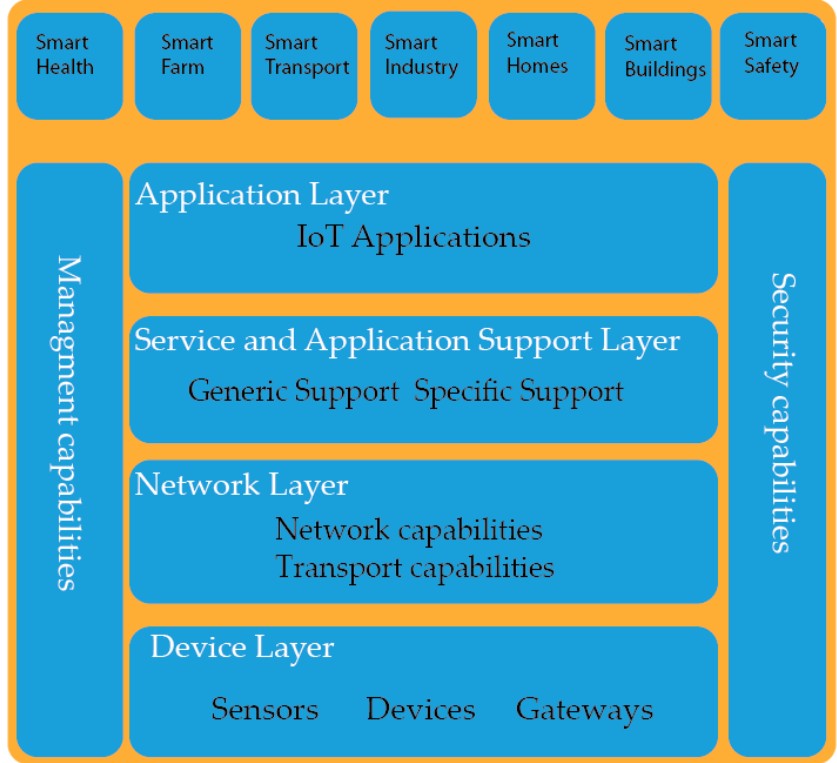

**Figure 5.** IoT generic model proposed by the International Union of Telecommunications (ITU) in recommendation Y.2060 [20].

IEEE and ISO have also proposed several standards related to smart city subcategories that cover aspect like interoperability, operation frequency, and security. Figure 6 shows an overview of such standard proposals.

Several IoT applications have been developed in the last four years using SIGFOX, NB-IoT, and LoRa technologies. These technologies were used because of aspects like easiness of configuration and low-cost hardware. LoRa is the focus of our work because it is an open-license technology and supports many smart city developments around the world. We carried out as SLR that allowed us to determine the key aspects of LoRa solutions related to different layers of the ITU architecture. The PRISMA methodology used in this work is divided in four stages: identification, screening, eligibility analysis, and inclusion. The identification stage included different steps such as study selection, inclusion and exclusion criteria, manual search, and removal of duplicates. The screening stage consisted in the review of titles and abstracts. The eligibility analysis stage was executed by reading the full texts of the selected articles. Finally, the inclusion stage consisted in the data extraction.

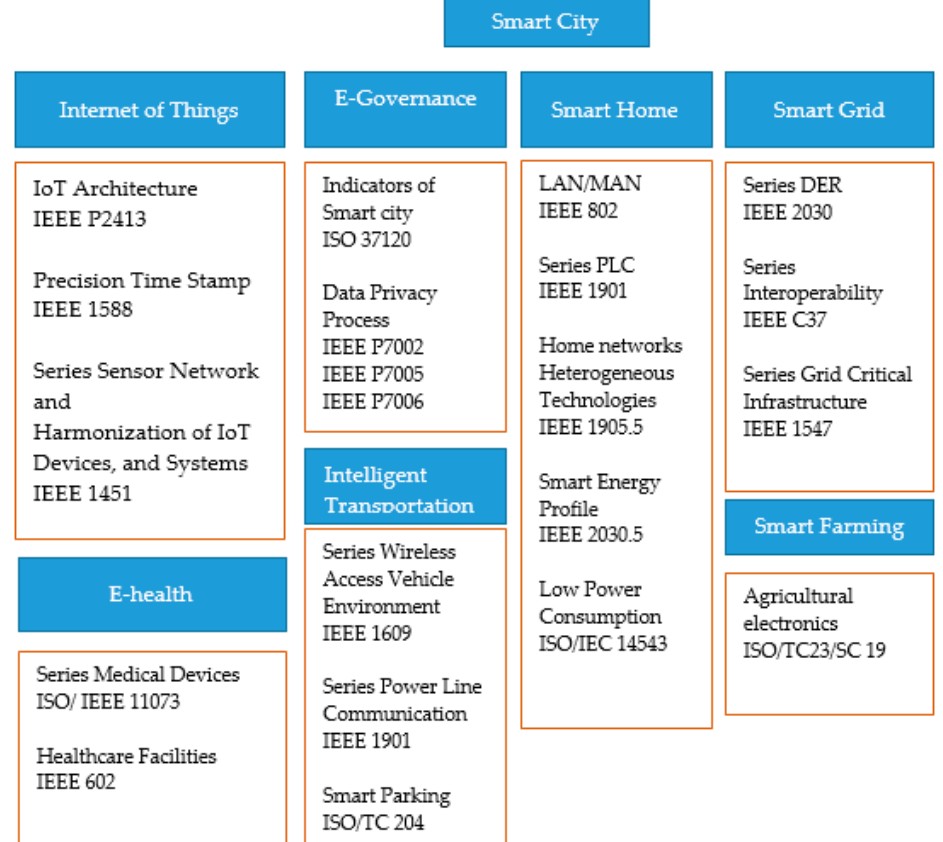

**Figure 6.** IoT standards for smart city subcategories. IEEE: Institute of Electrical and Electronics Engineers, ISO: International Standardization Organization.

### 3.1. Stage 1: Identification

#### 3.1.1. Study Selection

Study selection was based on a systematic review following the Prisma Guidelines [21]. It was executed using the following databases: Springer, Scopus, IEEE, Association for Computing Machinery (ACM), Web of Science, and Science Direct. These databases were chosen since they are the most relevant sources of information corresponding to Internet of Thing. All publications from 2015 to 2019 were included in the search, and the used the keyword string was "(IoT OR Internet of thing)" AND "(LoRa OR LoRAWAN OR LPWAN)".

#### 3.1.2. Inclusion and Exclusion Criteria

The inclusion criteria were: (i) documents published by peer-review academic sources; and (ii) documents that considered the use of LoRa for application development. On the other hand, the exclusion criteria included: (i) Preview surveys about LoRa (these works were included in the related works section but were not considered for the quality analysis process); (ii) Works that only included technical aspects like modulation, encryption, and protocols.

On the basis of the keyword string, we found 900 papers related to IoT and LoRa. Figure 7 indicates the searched papers distributed by year. It shows that 2018 was the year with the greatest number of publications, corresponding to 393 papers.

## Number of papers by year

**Figure 7.** Number of papers about LoRa and IoT from 2015 to 2019.

### 3.1.3. Manual Search

The identified 900 papers were 497 journals papers, 347 conference proceedings papers, and 56 book chapters. Table 3 shows the journals with the major number of papers related to IoT and LoRa from 2015 to 2019 according to our research.

**Table 3.** Number of papers in scientific journals from 2015 to 2019.

| Journal | Numbers of Papers |
| --- | --- |
| Sensors | 22 |
| Association for Computing Machinery International Conference | 17 |
| IEEE Internet of Things Journal | 12 |
| Ad Hoc Networks | 8 |
| IEEE Access | 8 |

### 3.1.4. Removal of Duplicates

Duplicates were removed through a manual review of the collected papers. In this process, we removed 126 papers.

### 3.2. Stage 2: Screening

#### Titles and Abstracts

We carried out a screening process of the 774 remaining papers for the selection of the main contributions. This process was based on papers' titles and abstracts using a web application created for the systematic review process. The web application allows each reviewer to see the titles and abstracts of the collected papers, while maintaining a blinded review process. The papers that did not comply with the inclusion criteria in the title or abstract were excluded in this phase of the study. At the end of the screening process, all reviewers could see the papers selected by the other reviewers. We resolved all disagreements among reviewers related to some excluded or included papers. At the end of this step, 116 articles fulfilling the criteria remained in the selected group.

*3.3. Stage 2: Eligibility Analysis*

Full-Text Reading

The full text of each article was examined, and the papers were selected if they were related to the development of applications using LoRa (including prototypes, implementation, or simulation). After this step, all 116 papers were selected for the next step. The different types of study selected are shown in Figure 8.

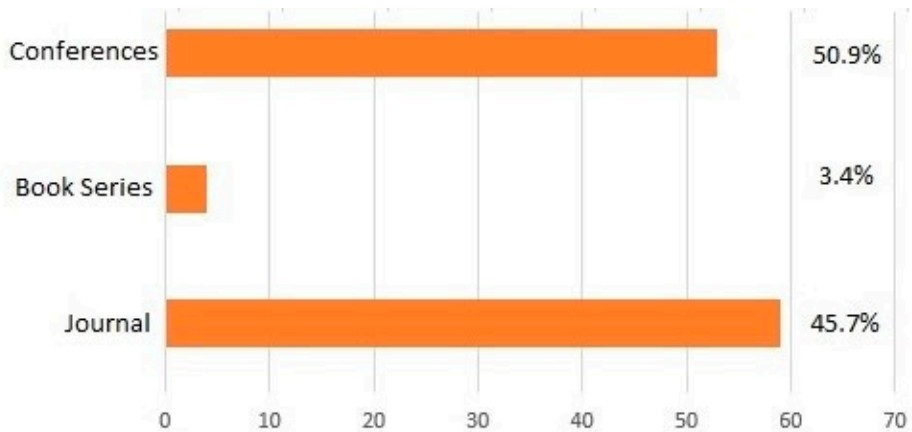

| Name | Type | Number |
|------|------|--------|
| Sensors | Journal | 25 |
| ACM International Conference | Conference | 17 |
| IEEE Internet of Things Journal | Journal | 12 |
| Lecture notes in Computer Science | Journal | 11 |
| IEEE Access | Journal | 8 |
| Ad Hoc Networks | Journal | 8 |
| IEEE International Symposium on Personal, Indoor and Mobile Radio Communications | Conference | 8 |
| Journal of Network and Computer Applications | Journal | 7 |

**Figure 8.** Type of papers about LoRa and IoT selected for the present study.

*3.4. Stage 4: Inclusion*

Data Extraction

For each selected paper, we summarized the following information: (i) type of application; (ii) data frame parameters; (iii) communication protocols; and (iv) coverage range and sensors types. For example, we identified 11 types of application (in parenthesis, the number of papers for that application is shown), i.e., Agriculture and Farming (13), Energy (10), Environment (23), Healthcare (13), Industry (13), Traffic (11), Waste Management (13).

The results of PRISMA methodology in this study are show in the Figure 9.

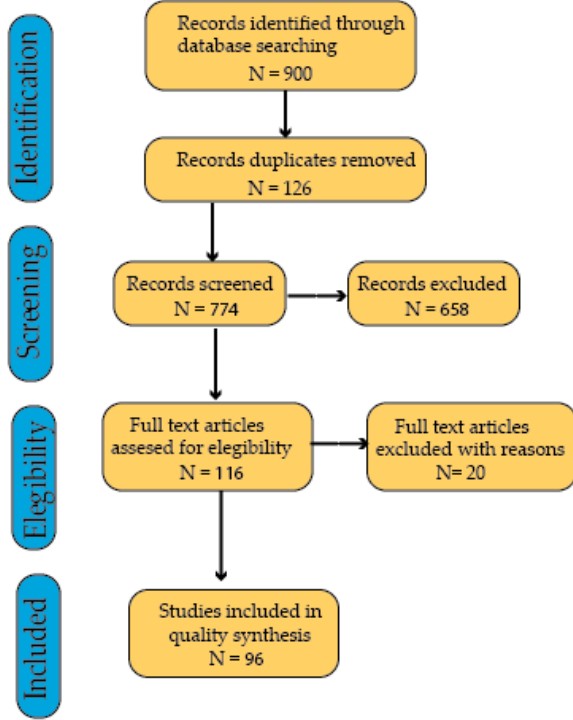

**Figure 9.** Methodology used for a systematic literature review and a qualitative analysis of LoRa-related papers.

## 4. Results of the Analysis of the Selected Works

### 4.1. Analysis of the Application Layer

The inclusion of ICT and sensors in different subcategories of the smart city such as agriculture, healthcare, environment, traffic, safety, and industry allows the creation of a strategic vision of the city. The use of LPWAN technologies is considered in smart city because it allows reaching large areas of coverage, while consuming a low amount of energy. Figure 10 shows a basic overview of our analysis of LoRa usage to support different kinds of applications, related to agriculture, healthcare, energy, industry, safety, environment, and traffic. The coverage of LoRa implementations identifies both urban and rural areas.

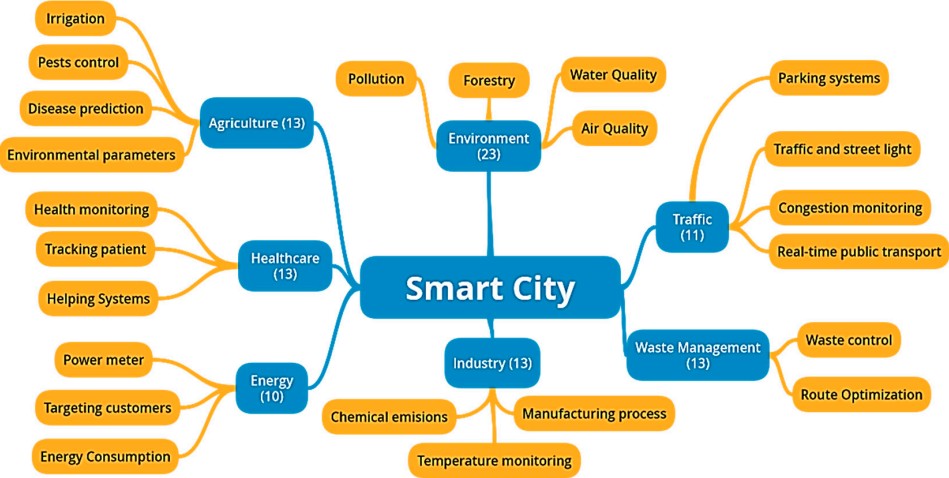

**Figure 10.** Types of LoRa applications in the smart city.

The use of IoT in different domains of the smart city establishes an ecosystem which includes: (1) technical aspects such as frequency of operation, energy levels, and sensitivity thresholds; (2) use of technical components such as sensors, imaging devices, and data capture; (3) use of data analytics processes based on cloud or fog infrastructures; and (4) parameters to be sensed and monitored according to the objective and particularity of each smart city domain. For instance, in the case of smart agriculture, parameters such as temperature range, humidity values, and soil shade are relevant to determine the feasibility of planting and harvesting. This data analytics process can be very useful for pest control in agriculture. According to our SLR, some components of the IoT ecosystem are used in the different subcategories of the smart city, for instance, fog or cloud computing are used for the storage and processing of the data collected by sensors in agriculture, transportation, or healthcare. Other components are more commonly used in some applications of specific subcategories of the smart city; for instance, biomarkers or wearables are most commonly used in healthcare applications. Complementary technologies like Radio Frequency Identification (RFID), Global Position System (GPS) and Drones are used to provide additional information related with geographical conditions. Figure 11 shows the most used components to create IoT ecosystems in smart city applications according to the qualitative analysis performed in this work.

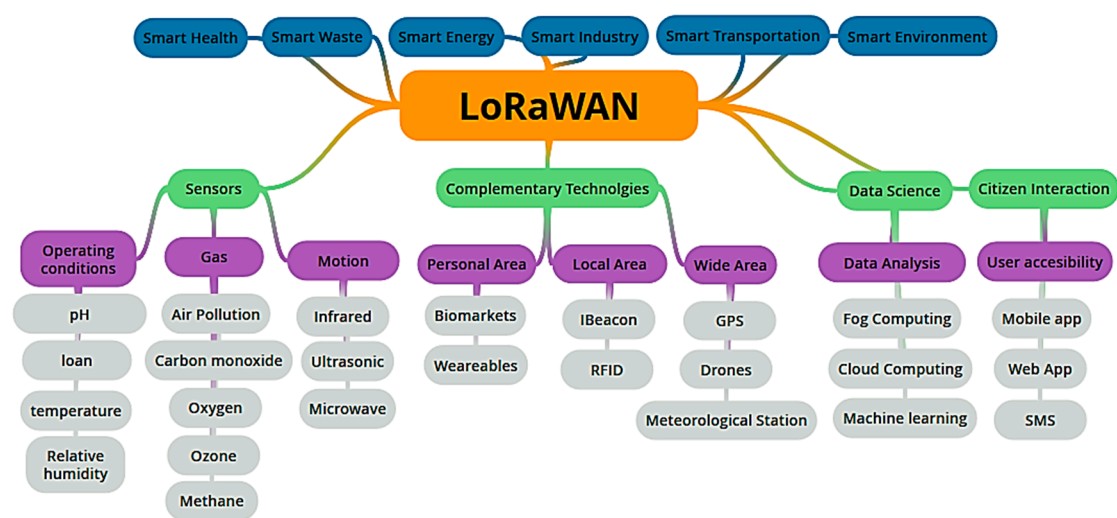

**Figure 11.** Components of IoT ecosystems in smart city applications. RFID:

According to our qualitative analysis, Lora-based IoT ecosystems for smart cities comprise five relevant components, i.e., sensors, complementary technologies, data science, citizen interaction, LoRaWAN.

The LoRaWAN allows the delivery of the data generated by different sensors and IT supplements to the data science component. Such data, through the data analysis process, are converted to patterns or anomalies, which are used by citizens and city administrators for adequate decision-making.

On the other hand, sensors allow obtaining parameter values that are relevant in each subcategory of the smart city. In this sense, we identified three groups of sensors that are the mostly used in IoT solution based on LoRa. The first group of sensors is focused on obtaining parameter values such as pH, loan, and temperature, which are related to operating conditions of certain city domains (e.g., agriculture, energy, and environment). For instance, in agriculture, the pH and temperature values allow the identification of conditions that are suitable for planting or harvesting. In water management, the pH value allows the evaluation of water quality. The second group of sensors includes those devices focused on the measurement of gases (e.g., air pollution, oxygen, and ozone). This type of sensors (according to our qualitative analysis) are greatly accepted especially in the environmental domain to evaluate the levels of pollution in a city. Finally, the third group includes those sensors used for motion detection. These types of sensor have had great momentum with the development of

valuable applications such as smart parking (sensors are used for the detection of available parking spaces) and security systems in smart home solutions.

The second component, which we have called "Complementary Technologies", considers those devices used to obtain additional data for specific applications. In this analysis, we identified three groups of elements. The first one includes local-area coverage devices that allow the determination of patterns of mobility of people through the use of iBeacon. The second group includes those devices of personal-area coverage that allow getting values of physiological parameters of people, such as breath rate and blood pressure. In this group, elements related to wearable devices and biomarkers are included. The third group includes devices of wide area coverage that allow increasing the sources of information to complement the data obtained by IoT devices. For example, the use of drones allows a system to get geospatial images of cultivation areas that, together with the pH and temperature values, would allow determining the cultivation conditions with more accuracy.

The third component, called data science, considers the techniques used for analyzing the data obtained from the other components. It includes the machine learning technologies used for determining patterns of behaviors (classified as normal or abnormal). It gives to the citizens or city managers an extra perspective for taking decisions. In the case of agriculture, the detection of an anomalous behavior could be related to a possible plague that affects crops. The data generated by IoT devices, sensors, and IT supplementary devices are large, so it is necessary to use cloud or fog computing platforms to store and process them. Since sending all the data to cloud platforms can be expensive or can generate delays, the use of fog computing platforms has grown, becoming more relevant in recent years.

The fourth component is focused on the accessibility of data for citizens and city managers. IoT solutions must generate information for improving the decision-making process. Therefore, it is important to deliver the appropriate information to the concerned people using, e.g., SMSs, mobile applications, and web portals. The way of presenting the information must consider aspects of user experience and accessibility.

Figure 12 shows an example of a system. In this example, sensors use a Local ID for identification and the information gathered by sensors are sent to the IoT gateway (GW) in fixed intervals of time. The IoT GW receives the Local ID sensor and sensed values which are redirected to FaaS, Fog, or Cloud computing infrastructures. The values sent by the sensors are stored in local database for data analysis. Then, based on the result of the analysis, control messages are sent to the actuators [18].

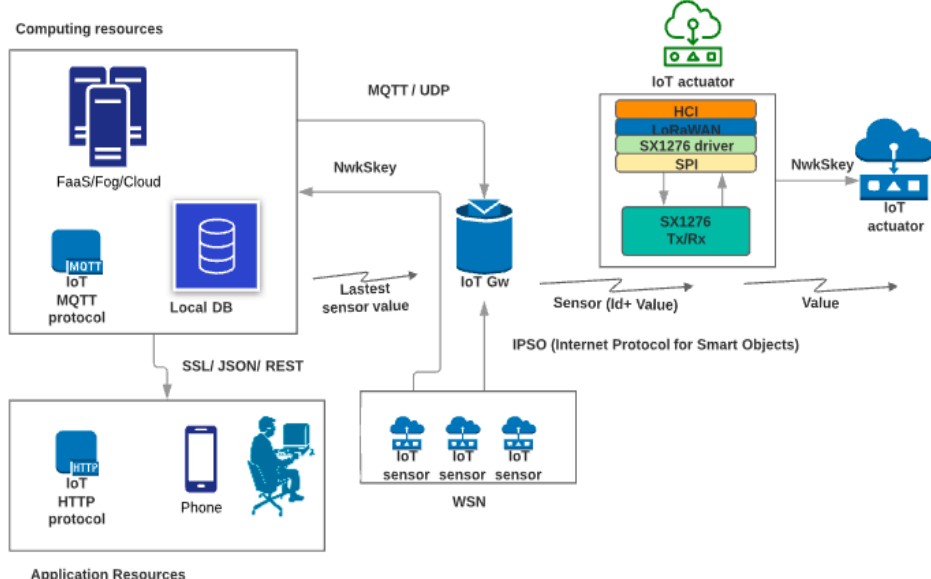

**Figure 12.** Technological components of an IoT ecosystem.

For an easy expansion and optimization of communication, a lightweight protocol is used. The most popular are the Message Queuing Telemetry Transport (MQTT) and the constrained application protocol (COAP), which are both designed for resource-constrained devices. MQTT supports multiple nodes communicating with a central broker. Information from the broker can be accessed in real time by the user using PCs or mobile devices [22].

The following subsections describe the operation process between the different components of the IoT ecosystem in relation to smart city subcategories.

### 4.1.1. Agriculture

IoT supports the process of data analytics providing environmental parameters such as humidity and temperature that are important for the life of plants and animals [22]. Variations in temperature and humidity are associated with climate changes and can lead to the appearance of pests or diseases. The size and shape of the sensors used in agriculture depend on the characteristics of the plants examined. They can be expensive, e.g., those used in large farms, and the information they provide can be enhanced by complementary data such as geo-space images [23]. The data generated by sensors can be processed by cloud or edge computing [24]. Additionally, it is possible to determine adverse effects on the plantation by using mathematical algorithms as time series [25].

The agriculture sector faces enormous challenges due to the need of producing food for the world's population, which will reach 9.6 billion people by 2050, according to the Food and Agriculture Organization (FAO). This means that the increase in food production must grow by 70% in the next years [26]. One alternative to address this challenge is improving the quality and quantity of the agricultural production. For this, the automation and continuous monitoring of pest and diseases as well as the identification of optimum weather and geographic field parameters would be very useful. IoT solutions integrated with the use of drones and 3D mapping allow the collection of information about the geography of a field and the soil composition to plan the plantation process [27]. Drones monitoring the status of crops and farms in real time facilitate the access of data coming from dangerous areas. In many cases, sensor nodes are installed in the farm, and the gateway node is carried by a drone to gather information from the sensors; then, the data are sent to servers, fog or cloud infrastructures, for processing.

Smart irrigation is another relevant application in agriculture that allows utilizing water efficiently at the appropriate moment and in the right amount. The gateway node of an IoT architecture can send commands to irrigation nodes to activate or deactivate a solenoid valve for starting or stopping the irrigation process [28]. The use of IoT solutions to determine the optimal water temperature and salinity allows taking actions for increasing the survival rate of larvae and adult crabs. Another IoT application in agriculture regards water quality monitoring. It is based on sensors that determine water pH, temperature, and salinity and deliver such information to the farmer in an intuitive way [29]. For forestry, it is important to find methods for the automatic monitoring of parameters like temperature, humidity, noise, solar radiation, pressure, soil, and moisture. Applications in forestry present major problems concerning the Fresnel, zone due to obstacles [30].

### 4.1.2. Healthcare

IoT applications in healthcare are focused on the monitoring of physiological and environmental aspects by the user to determine his/her health status [31]. The ubiquitous characteristic of IoT allows healthcare solutions for the follow-up of patients, especially those in critical conditions such as people with mental disorders [32]. IoT in healthcare also allows the development of systems that support people with physical limitations and those who have chronic diseases [33]. In the field of health, the monitoring process is intended to be the least invasive for people. To this aim, the use of wearables is considered [34]. The data generated by the different components can be processed in Fog computing [35]. The use of Cloud computing is considered in healthcare applications for processing

and storage, but in some cases, privacy issues have risen, since the information managed by the systems is very sensible.

### 4.1.3. Traffic Control and Transportation

IoT applications related to traffic control or transportation are mainly focused on parking systems, traffic, streetlights, congestion monitoring, and real-time public transportation. Generally, traffic in the center of a city is an important issue, not only because it increases the time of transportation for the citizens, but also because it decreases the gross domestic product (GDP) [34]. Smart lights are used to improve power efficiency and to reduce carbon dioxide emissions. Led lights consume approximately 0.49 MWH per led lamp, but the use of IoT allows the development of automatic dimming controls, improving the power efficiency of these light systems by over 85% [36]; dimming controls interact with passive infrared sensors that detect the presence of people and vehicles. Magnetometers allow the detection of objects like cars, motorcycle, and transportation vehicles. On the basis of data recorded from sensors at a traffic intersection, it is possible to design a congestion model to calculate the operating time of traffic lights for different lanes [37]. Finding empty parking spaces is another important issue for many cities, but the use of magnetic sensors is prone to deliver false positives. Three main challenges in the design of parking solutions based on magnetic sensors are (i) environmental changes; (ii) special characteristics of parking slots; and (iii) time-varying factors due to geomagnetic fields or dwindling battery lives [38]. Some researchers indicate that alternatives to sensing technologies such as RFID, ultrasonic sensors, or video–image processing require excessive overhead for installation and maintenance, while solutions that involve the detection of signals reflected from vehicles like microwave radars or ultrasonic sensors increase the power consumption and are extremely susceptible to environment interference. iBeacons and RFID are also used to determinate the available number of park slots [39].

Information about buses is an aspect considered relevant in the transportation domain because it delivers data for citizens about when a bus will arrive to a bus station. Bus information is sometimes difficult to collect in a precise manner due factors like weather and congestion. One solution of this issue is the usage of GPS for tracking bus location. However, it requires expensive communication charges, and the user needs a mobile network to access such information. An cheaper alternative could be LoRa [40]. A bus location information acquired by GPS would be broadcasted to bus stations using the LoRa network. This solution would reduce communication charges [41]. Other solutions that could be introduced using LoRa are air pollution and road surface condition monitoring [42].

The use of bikes for transportation in cities is growing as an alternative to reduce air pollution. In this sense, citizens can borrow a bike and return it to any bike station around a city. Sensors are installed in bikes for collecting real-time information about the routes where a bike was used. Additionally, the system could tell the user where a nearby bike station is. In contrast to traffic models developed for buses in which some movement patterns are expected, bikes networks are more unpredictable and complex to define, since each user will move in a different way. The integration of weather information could add new services for the user, allowing him/her to know if the weather conditions are comfortable for riding a bike [43].

### 4.1.4. Energy

Accuracy in electrical billing is highly needed, especially with the widespread use of smart-home appliances in the daily lives. The development of automatic meter reading (AMR) avoids the need for a person to visit a physical location for reading the meter and reduces the possibility of reading the data wrongly [44]. Currently, General Packet Radio Service (GPRS) solutions are used in this context, but they are quite expensive and can lose SMS packets. In this situation, the use of LoRa would allow the creation of a private network including a third-party service. To improve the electrical billing service, a two-way communication between users and providers could be implemented. This would allow the user to see the energy consumption in real time.

Another approach related to energy is the monitoring of power plants. With the growing of the worldwide population, fossil fuel consumption has been increasing in the last years. For this reason, photovoltaic (PV) power plants are considered as an alternative solution. Since the PV plants are highly dependent on solar irradiation, ambient temperature, and atmospheric conditions, IoT solutions could allow monitoring these parameters in real time [45].

### 4.1.5. Environment

Air pollution is the product of human activities executed daily, such as the use of transport vehicles and of appliances in residential and industrial settings. These activities can generate a big amount of carbon monoxide that affects the air quality. If pollution is uncontrolled, it could cause cancer, respiratory illnesses, and cardio-vascular diseases [46]. The World Health Organization has identified tolerable pollution level thresholds and has defined the concept of "atmospheric pollution" [43], which refers to the presence of airborne pollutants such as fuels, carbon, bacteria, and other microscopic particles in the atmosphere. Six pollutants, i.e., ozone ($O_3$), nitrogen dioxide ($NO_2$), sulfur dioxide ($SO_2$), carbon monoxide (CO), and two types of fine particulate matters (PM10, PM2.5) are detrimental to both environment and public health [46]. Table 4 shows the relation between pollutants and diseases. Air quality is directly related to people health and quality of life. Typically, people spend more than 90% of their time in indoor environments [47]. $CO_2$ levels over 1000 ppm indicate a potential air problem in indoor environments. In this sense, the continuous monitoring of $CO_2$ concentration is important to detect any anomaly in the quality of air.

**Table 4.** Diseases due to pollutants. PM: particulate matter.

| Pollutants | Health Disease |
|---|---|
| $NO_2$ | Lung damage |
| CO PM 2.5 PM10 | Respiratory and hearth diseases |
| $SO_2$ $O_3$ | Asthma |

The use of IoT solutions in ambient assisted living (AAL) permits the design of smart objects with the capabilities of sensing the environment and executing specific actions for their users. IoT sensors installed in the urban environment also allows the detection of pollution concentration in real time [48]. With the collected information by the aforementioned solutions in conjunction with machine learning algorithms and data analytics, it is possible to identify and generate warnings in case of high levels of air pollution.

### 4.1.6. Waste Management

The increasing population density generates more garbage every day, which increases the need for more physical space for waste and the associated costs to clean, bury, and burn it. In waste management, three important aspects to define are: (i) the collecting routes; (ii) an adequate waste classification system; and (iii) the management of the physical space for garbage. A waste management system can consider a compression device; when a garbage container is full, a step motor is activated to compress the garbage and save space [49]. Additionally, with the use of a variety of sensors, such as infrared, ultra-sonic, and methane sensors, it is possible to generate an automatic classification of garbage and distinguish between recyclable and non-recyclable garbage. The use of the RFID technology also allows the identification of waste to indicate the correct garbage bin where it should be deposited. An important aspect to be considered in this work is data confidentiality, since the garbage of a

person or family can contain sensitive information. Regarding this issue, the authors of some studies considered the usage of secure communication protocols such as https in their solutions [50].

Wastewater overflow monitoring is another important issue for the modern city, and consists, for example, in detecting possible problems in the flow of wastewater and in analyzing the effect of waste in sensitive areas. To control wastewater, IoT systems can uses radar sensor platforms, sewer pump stations, and LoRa to cover a large portion of a city [51]. With a single gateway, it is possible to collect the information of water-waste levels of sewer pump stations and then analyze the data to estimate the pump flow; this allow identifying when sewer pump stations are full, avoiding the overflow of wastewater.

### 4.2. Analysis of the Network and Transport Capabilities

#### 4.2.1. Data Layer

Data packets in LoRa can contain up to 255 bytes. This characteristic makes LoRa a low-speed data transfer technology, which is acceptable in sensors networks. Data frames in IoT applications can have different lengths which depend on different factors such as type of data, numbers of packets to be sent, design of the application, and delay requirements. According to our quality assessment, we identified IoT solutions including short packets of 8 bytes or long packets up to 128 bytes. For example, a smart light monitoring solution used 10-byte frames for communication among lights [52], an agriculture applications (monitoring of a star fruit plantation) used 5-byte-long frames [53], and a bike monitoring solution used 12-byte-long frames to determine routes for cyclers [54]. Additionally, a health application used 93-byte data frames to send health information such as systolic/diastolic data and pulse [34]. In the case of water quality, information about salinity and temperature was sent in a 17-byte data frame [55]. The variable data frame length of LoRa provides greater flexibility in the development of IoT applications for smart cities.

The length of the data frame includes payload and header, and this length can impact on the transmission time and energy consumption. Technical factors that can influence the data length are BW, SF, and throughput. On the other hand, LoRa uses the chirp protocol. The "up-chirp" moves upward in frequency, and the "down chirp" moves below the central frequency; the central frequency depends of the geographic region and is 915 MHz in the USA (also frequencies in the range between 902 MHz to 928 MHz), 868 MHz in Europe (planned 433 MHz), 470 MHz in China, 923 MHz in Japan, 915 MHz in Australia (also frequencies between 915 MHz to 928 MHz), and 433 MHz in the rest of Asia. LoRa uses three bandwidths: 125 KHz, 250 KHz, and 500 KHz, and SF, that is the duration of the chirp, can be SF7 for the shortest time in air and SF 12 for the longest time in air [56].

Tables 5 and 6 and Figure 13 show the relation among data rate, frequency, throughput, and payload in LoRa technology. If the selected SF is high, then throughput and payload are low. For example, for the same bandwidth of 125 KHz, in the frequency of 868 MHz, Table 4 indicates that if the selected SF is 7, the throughput is 5470 bps, and the payload is 230 bytes. However, if the selected SF is 12, then the throughput decreases to 250 bps, and the payload is reduced to 59 bytes. Bandwidth selection also affects the length of the payload. For example, in Table 6, for the same frequency of 915 MHz and the same SF of 7, if the bandwidth selected is 500 KHz, the throughput is 21,900 bps, and the payload is 230 bytes; in contrast, if the selected bandwidth is 125 KHz, the throughput decreases to 5470 bps, and the payload increases to 250 bytes.

**Table 5.** LoRa parameters in EU868, EU433, CN470, and AS923 bands, frequency of 868 MHz [56].

| Data Rate | Spread Factor | Bandwidth (KHz) | Throughput (bps) | Payload (Bytes) |
|---|---|---|---|---|
| DR6 | SF7 | 250 | 11,000 | 230 |
| DR5 | SF7 | 125 | 5470 | 230 |
| DR4 | SF8 | 125 | 3125 | 230 |
| DR3 | SF9 | 125 | 1760 | 123 |
| DR2 | SF10 | 125 | 980 | 59 |
| DR1 | SF11 | 125 | 440 | 59 |
| DR0 | SF12 | 125 | 250 | 59 |

**Table 6.** LoRa parameters in US902-928 and AU915-928 bands, frequency of 915 MHz [56].

| Data Rate | Spread Factor | Bandwidth (KHz) | Throughput (bps) | Payload (Bytes) |
|---|---|---|---|---|
| DR13 | SF7 | 500 | 21,900 | 230 |
| DR12 | SF8 | 500 | 12,500 | 230 |
| DR11 | SF9 | 500 | 7000 | 230 |
| DR10 | SF10 | 500 | 3900 | 230 |
| DR9 | SF11 | 500 | 1760 | 117 |
| DR8 | SF12 | 500 | 980 | 41 |
| DR4 | SF8 | 500 | 12,500 | 250 |
| DR3 | SF7 | 125 | 5470 | 250 |

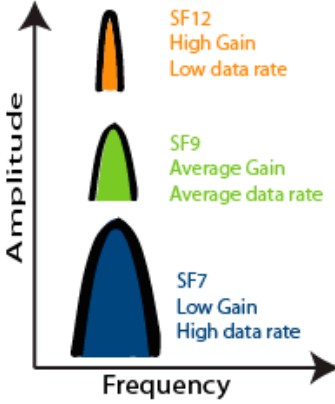

**Figure 13.** Relation among data frame, spread factor (SF), and bandwidth [56].

### 4.2.2. Coverage of LoRa in IoT Applications

LoRa's coverage can be measured on the basis of different parameters, such as received signal strength level (RSSI), signal-to-noise ratio (SNR), and packet delivery ate (PDR). SNR is an indicator of the link quality with respect to a radio channel´s environmental parameters [57]. IoT applications in smart city require different coverage areas depending on the type applications, location of the sensors, and type of communication, i.e., point-to-point or central-gateway connections. These aspects are mainly associated with the design requirements of the application. Table 7 presents an overview of the type of coverage for each subcategory of smart city applications. On the basis of our quality assessment, healthcare applications consider a fixed communication between a diagnostic system and a base station, but there are also some applications, e.g., those focused on patient follow-up, which require signal quality during communication when the objects are in movement (the speed of a patient can be in the ranges of 1–3 km/h, 5–8 km/h, and 1–30 km/h) [58]. In the case of traffic control or transportation applications, the monitoring of streetlight systems requires communication among multiple nodes located within a short range. The optimal distance between streets light is around of 35 meters [52]. The RSSI value can be used to determine the location of the user in parking systems through the use of iBeacon [40]. If the RSSI value is high, then the user is considered to be close to a parking slot.

**Table 7.** Localization of sensors in smart applications.

| Subcategory | Indoor | Outdoor | Object Motion |
|---|:---:|:---:|:---:|
| Agriculture | ● | ● | |
| Healthcare | ● | | ● |
| Energy | ● | ● | |
| Waste Management | ● | ● | |
| Traffic and Transport | | ● | ● |

Table 8 shows an overview of signals parameters related to IoT applications for different smart city subcategories. In agriculture, the communication is generally between server and nodes, and the nodes are commonly fixed objects. These types of applications require to consider different issues such as obstacles, foliage of trees, geography of the field, and climate conditions. Healthcare applications require communication between diagnostic nodes and a base station. Both objects in this scenario are at a fixed location. In other cases like tracking of patients, it is necessary to communicate with objects in movement. In this scenario, multipath fading and Doppler effects are more relevant. In traffic applications, RSSI is used for detecting parking slot occupancy. The difference between RSSI values is used to know when a vehicle is arriving at or leaving a spot. The RSSI changes are more pronounced in nearby slots. An RSSI exceeding $-90$ dBm is an indication of the occurrence of a transient event.

**Table 8.** Signal parameters (received signal strength level (RSSI), signal-to-noise ratio (SNR), packet delivery rate (PDR)) of subcategories of the smart city.

| Subcategory | Communication | Signal Parameters | Observations | Ref |
|---|---|---|---|---|
| Agriculture | Server–Node | @100 m, RSSI = $-69$ dBm, PDR = 100%, SNR = 6 dB @400 m, RSSI = $-100$ dBm, PDR < 80%, SNR = 3 dB @700 m, RSSI = $-120$ dBm, PDR < 41%, SNR = 1 | Decrease of parameters values due to distance and obstacles, e.g., trees. | [59] |
| Healthcare | Diagnostic System–Base station | @1.1 km, RSSI = $-127$ dBm, PDR = 100%, SNR = $-17.2$ dB @3.8 km, RSSI = $-121$ dBm, PDR < 80%, SNR = $-16$ dB @5.5 km, RSSI = $-109$ dBm, PDR < 41%, SNR = 1.1 dB | LoRa transmission can generate delays in sending information. Therefore, it is not recommended in real-time applications. Movement of patients can be in low-coverage areas, but it can cause loss of the connectivity signals. | [57] |
| | Tracking movement | RSSI = $-83.83$ dB, | | [60] |
| Traffic | Streetlights | @50 m, RSSI = $-34$ dBm, PDR = 100%, SNR = $-17.2$ dB @250 m, RSSI = $-110$ dBm, PDR = 100%, SNR = $-17.2$ dB | Optimal distance between streetlights is around 35 m. This situation requires a large number of sensors. | [37] |
| | iBeacon user localization | RSSI = $-95$ @ $-99$ dBm | An abnormal RSSI value is <$-70$ dBm. This situation allows to identify empty slots or determine user location. | [46] |
| | Parking slot occupancy | RSSI = $-80$ dBm | Difference between empty and occupied slots is near 12 dB. Transient events >$-90$ dBm | [39] |

Indoor coverage values higher than $-70$ dBm are considered acceptable. On the other hand, for outdoor coverage, the power of LoRa signals can be affected by obstacles on the line of sight; in this scenarios, values higher than $-100$ dBm are considered of good quality [31]. The location of the LoRa gateway in indoor or outdoor scenarios can impact the coverage area [61].

*4.3. Analysis of the Device Layer*

The main objective of the communications infrastructure is to interconnect nodes and generate a way to access the data generated in different domains of the city.

4.3.1. Sensors

Sensorization allows abstracting data from the physical world's to the digital world. The continuous monitoring of physical variables like temperature, humidity, and pH allow users to take decisions about specific aspects of smart city domains. Table 9 and Figure 14 present an overview of the sensors

used in smart city subcategories, based on our quality assessment. In the agriculture subcategory, some environmental parameters such as humidity, temperature, and luminosity must be measured to predict climate changes [22]. This detection can be based on a single-sensor or on a multi-sensor platform collecting microclimate data [23]. Magnetic sensors can be used in smart parking applications to detect empty park slots. This type of sensors can be influenced by nearby ferrous metal objects like bicycles or adjacent vehicles. The earth's magnetic field can also alter the data generated by this type of sensors [39]. Light sensors allow the determination of the amount of light. This is useful in plantations as well as in AAL or smart homes to turn on the lights in the night. Light sensors can be affected by outdoor lighting conditions like clouds or stationary objects that block the sun light. Dust sensors are based on the light-scattering detection technology that estimates the concentration of particles that are suspended in the air. Through the evaluation of the amount of light that is backscattered by the particles, it is possible to determine the air quality [62]. Some aspects that are important when using sensors are (i) timestamp and (ii) sensor identification. The timestamp between a node and the gateway must be synchronized; this allows validation of the information received by the nodes as new or old [25]. For sensor identification, a Local ID could be used. This is useful for knowing which device is sending the information.

**Table 9.** Types of sensors used in different subcategories of smart city applications.

| Subcategory | Sensing Parameters | Type of Sensors | Distance GW-Sensor | Ref |
|---|---|---|---|---|
| Agriculture | Humidity, Temperature, Luminosity, Solar radiation, Soil, Conductivity, Ph | Ultrasonic Temperature Humidity Soil | 30 cm–15 km | [22,63–66] |
| Healthcare | Health signs | Biosensors | 3 m | [61] |
| Energy | Light intensity Motion Voltage Temperature Humidity | Temperature Humidity Motion | 15 km | [18,37] |
| Traffic | Motion Occupancy | Magnetic Ultrasonic | 500 m–1 km | [41,42] |
| Environment | $CO_2$, $NO_2$, $O_3$ Concentration Weather | Gas Temperature | 200 cm–5 km | [27,67,68] |

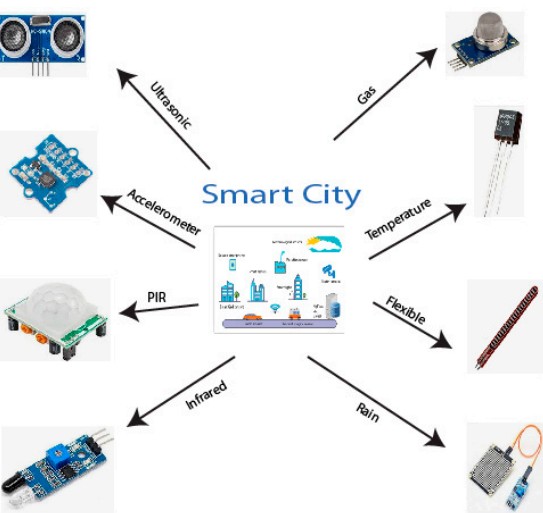

**Figure 14.** Sensors used in smart city applications.

The power consumption of sensors varies depending on their type. Although the consumptions are low (voltage ranges between 2.2 to 5 V and amperage is less than 1 mA), it is necessary to consider energy consumption in the design of IoT solutions, since the end nodes depend on low-powered lithium batteries. Table 10 shows the most common voltage and amperage ranges used by sensors.

**Table 10.** Generic specifications of sensors.

| Subcategory | Range |
|---|---|
| Voltage | 2.2 V to 5 V |
| Amperes | Temperature sensor: 0.84 mA<br>Light sensor: 0.56 mA<br>Accelerometer: 4.68 mA<br>PIR: 0.75 mA<br>Gas sensor: 4 mA–20 mA<br>Infrared sensor: 5.5 mA<br>Flex Sensor: 20 mA |
| Standards | IEC-61724, IEEE 1451 |

On the basis of the SLR carried out in this work, we could identify several IoT–LoRa projects using DTH XX series sensors due to their easiness of configuration, low cost and good accuracy. Additionally, the availability of libraries for this type of sensors in different programming languages allows time and effort reduction for the development and implementation of LoRa-based IoT solutions. The sensor DTH11 and DTH 22 are digital sensors; DTH22 allows temperature measurements in the range from −40 to 80 °C with accuracy of ±0.5 °C and humidity measurements in the range from 0% to 100% RH (relative humidity) with an accuracy of 2% RH. DTH11 works in a temperature measuring range from 0 to 50 °C with an accuracy of ±2.0 °C and in a humidity range from 20% to 90% with an accuracy of 4% RH.

The sensors DTH11 and DTH22 can transmit bits to the nodes of LoRa ecosystems. Table 11 shows the bits sent in different group of bytes [64–66]. A complete transmission of information from DTH 11 and DTH 22 consists of 40 bits (5 bytes) in a period of time of 4 ms. Two bytes correspond to the temperature measurement, two bytes to the humidity measurement, and 1 byte to errors checksum.

**Table 11.** Types of sensors for smart city applications.

| Byte | Information into Bits Groups |
|---|---|
| First Byte | The entire part of relative humidity. |
| Second Byte | The decimal part of relative humidity. |
| Third Byte | The entire part of temperature. |
| Fourth Byte | The decimal part of temperature. |
| Fifth Byte | Checksum of all previous bytes. |

DTH11 and DTH 22 are not high-precision sensors, but their accuracy is acceptable. DTH11 can be used in tests or home projects, while DTH22 operates with medium accuracy and can be used in monitoring implementations at the city level. Table 12 shows a comparison of these two types of sensors based on their level of accuracy.

Some points to consider related to the connection between sensors and nodes are that values lower than 3.3 V can present problems, the power of the nodes must consider the limited capacities of the batteries, and the distance between sensors and nodes must be taken into account.

**Table 12.** Comparison of temperature sensors used in smart city projects.

| DTH11 | DTH22 |
|---|---|
| Temperature measured between 0 and 50 °C | Temperature measured between −40 and 125 °C |
| Temperature measure accuracy 2 °C | Temperature measure accuracy 0.5 °C |
| Humidity measured 20% to 80% | Humidity measured 0% to 100% |
| Humidity measure accuracy 5% | Humidity measure accuracy 2–5% |
| Samples frequency 1 Hz | Samples frequency 2 Hz |
| Voltage 3.5 V a 5 V | Voltage 3.3 V a 6 V |
| Amperes consumed 2.5 mA | Amperes consumed 0.3 mA |

### 4.3.2. Nodes

Nodes receive and send information to a gateway. Nodes can communicate in two ways within a LoRa ecosystem:

- Point to point: the node does not require an intermediary device and can communicate directly with other nodes.
- Mesh: a gateway is in charge of coordinating the communication between nodes in the network. It has a limited capacity of 250 nodes.

LoRa usually communicates in two directions. It can also broadcast to all nodes [69]. Figure 15 shows LoRa class nodes on the basis of LoRa MAC Layer operation:

- Class A: Offers greater energy savings. It is in listening mode after sending data to the gateway.
- Class B: Nodes have default reception windows with the gateway
- Class C: Presents the lowest energy savings

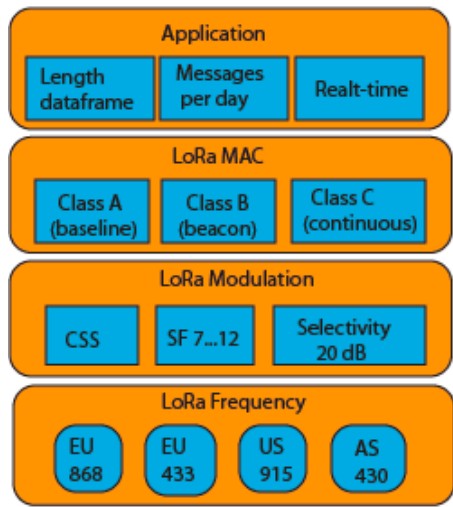

**Figure 15.** LoRa class on the basis of LoRa MAC layer operation.

Nodes must not only measure the required parameters in the different contexts of smart cities, but also need to send the data collected in real time. Depending on the type of application in the city, nodes may require a set of sensors. For instance, to measure nitrate concentration levels, the node needs to manage an array of gas sensors including $CO_2$, NOx, and $O_2$ sensors [70]. In some cases, nodes can have different roles. For example, in eHealth, there are safe nodes and health nodes which collect environmental and physiological characteristics. In agriculture, nodes are structured on the basis of roles, i.e., as measuring and receiving nodes [71]. The first ones collect data, the second ones execute some specific action. In projects of measurement of nitrate levels, one node is a sensing node and is in charge of collecting water samples from a stream or lake and measuring nitrate concentrations [63].

In order to be able to sense a specific aspect of the city, several nodes may be required. For instance, in smart farming, to determine the presence of castors, about 36 nodes are necessary to cover four locations [72]. In some deployments, thousands of nodes can be implemented in a LoRa network [73]. This is an advantage of LoRa; it can support multiple nodes, consuming little transmission power and allowing the delivery of frames with few bytes. Nodes are allowed to transmit up to14 dBm in Europe and up to 22 dBm in America [74]. For transmission, nodes use a transceiver for communication [75,76].

According to our literature review, different modules are used for the communication between nodes, for example, ESP8266, SX1272, and SX 1276. ESP8266 is an integrated low-consumption chip with 17 General Purpose Input Output (GPIO) ports that allow incorporating LoRa transceiver modules. The operating voltage of ESP8266 is in the average range between 3 and 3.6 V, and the operating current is 80 mA. The operating temperature range is from −40 to 125 °C. The main communication buses used by ESP8266 are SPI, I2C, UART, and the module supports protocols such as TCP, UDP, HTTP, FTP.

ESP8266 uses three operation modes:

- Active mode: At its full capacity, it consumes about 170 mA.
- Sleep mode: A real-time clock (RTC) is active for synchronization. In this mode, ESP8266 maintains data connection and does not require re-establishing the connection again. In this mode, the chip consumes between 0.6 mA and 1 mA.
- Deep Sleep: The RTC is not operational. Unsaved data are lost. In this mode, the device consumes about 20 uA.

The SX1272 provides ultra-long-range spread spectrum communication and high interference immunity. The operation frequency is 860–1000 MHz. It has other characteristics such as 157 dB as the maximum link budget, programmable bit rate up to 300 kbps, high sensitivity down to −137 dBm, modulation types FSK, GFSK, MSK, GMSK, LoRa, and OOK, and 127 dB dynamic-range RSSI. The SX1276 transceivers provide ultra-long-range spread spectrum communication and high interference immunity. The frequency of operation is between 137 and 1020 MHz, and its maximum link budget is 168 dB. Its programmable bit rate is up to 300 kbps, its high sensitivity is down to −148 dBm, and the used modulation types are FSK, GFSK, MSK, GMSK, LoRa, and OOK [77].

Several commercial nodes are available in the market. However, according to our literature review, smart city applications can be developed using low-cost hardware such as Arduino and Raspberry pi, which maintain an adequate level of performance and accuracy. Figure 16 shows some of the Arduino models used in LoRa implementations, i.e., Arduino Nano, Arduino Mega, Arduino Uno, and Raspberry pi model B. The Arduino hardware allows connecting different classes of sensors and transceiver communication modules. The main characteristics of the Arduino models is the number of GPIO and their cost. The design of IoT solutions should consider this aspect in the development phase [78].

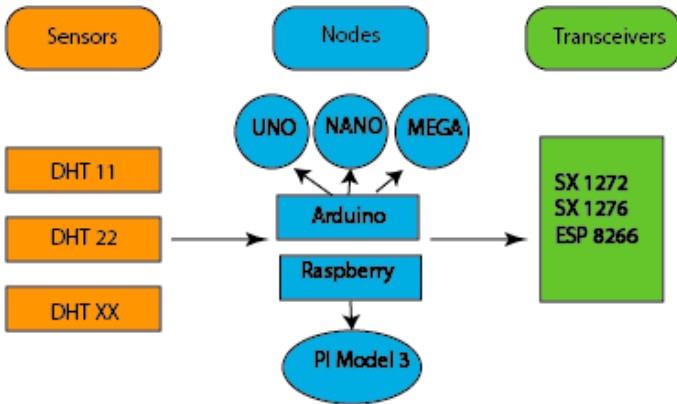

**Figure 16.** Sensors, end-nodes, and transceivers used in LoRa ecosystems.

## 5. Discussion

### 5.1. Smart City Strategies Using IoT–LoRa

To become a smart city, more sustainable and socially inclusive, a city should take advantage of IoT technologies. Several cities around the world are using LoRa to transform and improve their services in different areas such as health, education, and transportation. Amsterdam was the first city covered by LoRa using only 10 gateways. Other cities in other continents followed. Figure 17 shows some cities covered by LoRa, which include North American cities such as Boston, Ottawa, and Mexico City, South American cities such as Sao Paulo, Montevideo, and Buenos Aires, and Asian and European cities such as Pekin, Manchester, and Zurich.

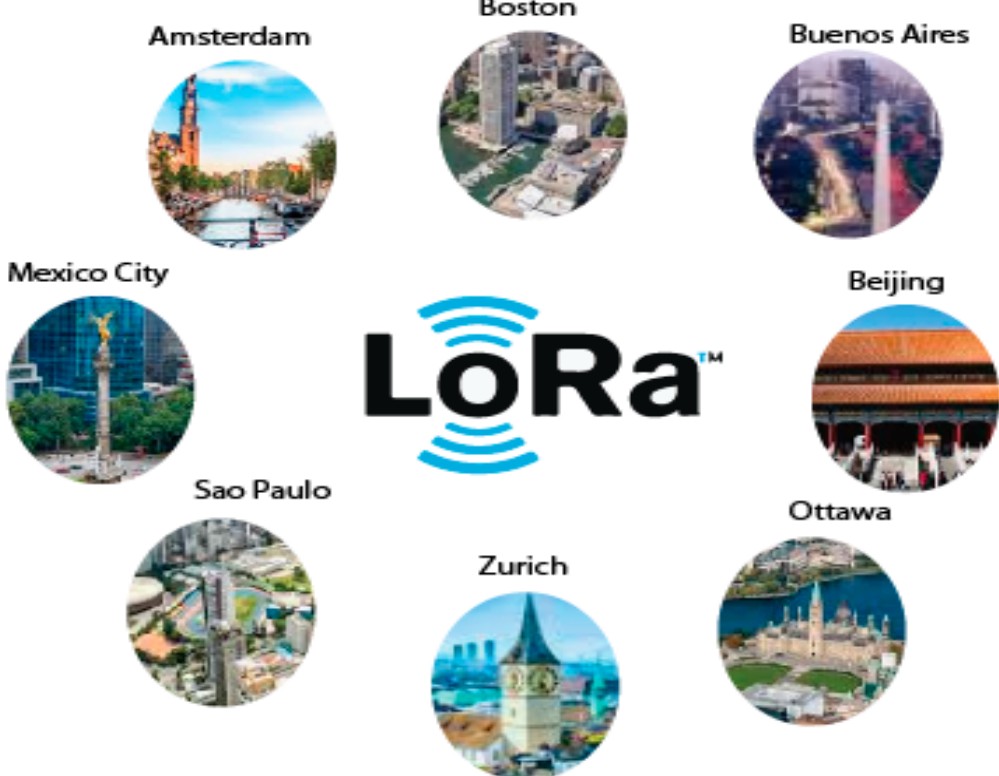

**Figure 17.** Smart cities according to ITU in 2019 [79].

Manchester has invested in transportation, social healthcare, culture, energy, and environment, considering that the data deliver information to allow the definition of better policies and create a more dynamic city infrastructure. Through sensors, it is possible to improve different types of services for visitors and citizens travelling in a city. For example, cyclists can detect when they are approaching a dangerous road; they can also detect road conditions and receive alerts in case of accidents. Manchester used six LoRa servers and 25 base stations to cover the city. Montevideo is another city using LoRa. This city has focused on the management of public lighting, water meters, noise measurement, agricultural services, and optimization of garbage containers. On the other hand, Zurich uses IoT to determine places with forgotten bikes, and zones with distribution grid power failures. Table 13 shows a short description of IoT applications used in some smart city strategies.

**Table 13.** Short description of IoT applications in smart cities.

| Smart City | IoT Applications in Smart Cities |
| --- | --- |
| Buenos Aires | Buenos Aires uses IoT to monitor parameters such as energy consumption, water level in reservoirs, and environmental conditions in 40 schools in the city. Another project is focused on measuring the flow of rivers to avoid flooding in nearby towns. Additionally, Buenos Aires makes use of IoT to create a scalable lighting system to make the city safer, more sustainable, and energy-efficient and to reduce light pollution [80,81]. |
| Amsterdam | Amsterdam has developed projects that use IoT focused on solving traffic-related problems by installing sensors that monitor the state of traffic flow and parking availability. Such solutions have reduced the time of searching available parking lots by 43% [82]. |
| Sao Paulo | Brazil has 400,000 connected devices and invests around 25 percent of Brazilian GDP. Several cities in Brazil, including Sao Paulo, have promoted the development of applications for the measurement of electrical energy, irrigation systems for farms, and sensorization of street lighting systems [83]. |
| Zurich | Zurich considers the use of IoT-LoRa for projects such as (a) no bike left (to recover abandoned bikes), (b) detection of power distribution grid failure, and (c) real-time occupancy rate monitoring in public transportation [84]. |
| Ottawa | In 2009, the city of Ottawa launched a digital storefront using a sensor structure. In addition, the city works with the Centre for Excellence in Next-Generation Networks (CENGN) to accelerate the inclusion of ICT in the city. CENGN provides an LPWAN infrastructure (i.e., an antenna and a gateway) to develop smart health, smart farming, and other applications [85,86]. |
| Beijing | Beijing invests around $5 billion in IoT projects related to transportation and environmental management. Beijing's Palace Museum uses IoT solutions to detect the movement of relics that may be associated with a robbery. Another problem that the city of Beijing seeks to solve with the use of IoT and cognitive computing is the reduction of smoke in the environment by installing several sensors to determine the sources of pollution [87]. |
| Boston | Boston works on smart city projects related to autonomous vehicles, intelligent parking lots, and interactive public art. The city uses cameras and sensors to learn how people navigate and interact with the city streets of Boston [88]. |
| Mexico | Mexico City seeks to strengthen the domains of governance, public management, and transportation. Mexico City tries to solve transportation and traffic problems by determining car flow patterns and delivering information of available parking spaces [89]. |

## 5.2. Social Contribution

Smart cities promote spaces of participation for citizens and establish communication channels among citizens, city administration (e.g., medical, transportation, and energy departments), and city services (e.g., parking, traffic management, and welfare systems). The interaction city–citizen can be improved with IoT–LoRa solutions. This type of technology inclusion promotes new dynamics in the social behavior of people. In the context of energy, citizens could find alternatives to reduce their energy bills. To this goal, research proposals are focused on designing smart home solutions. Smart home solutions allow tracking electrical devices and controlling their energy consumption. In the environment subcategory, the information generated by IoT systems could be shown in open-data portals. In public transportation, citizens could know the waiting times for a transportation or route changes. In the subcategory of waste management, citizens could monitor the level of waste and know the availability of waste recollection systems.

### 5.3. Industrial Contribution

The inclusion of IoT in smart cities creates value for taking decisions about different aspects of the city. With the literature review conducted in this study, we have identified five trends of IoT solutions implementation in smart cities, as shown in Figure 18.

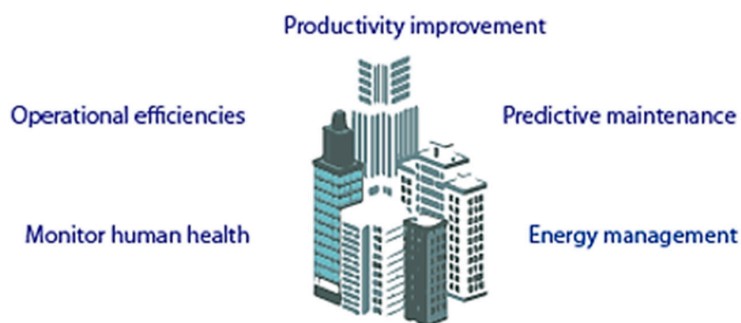

**Figure 18.** Trends of IoT solutions in smart cities.

Each city has its own strategy to become a smart city and has its own requirements for the different areas of services, such as health, environment, and transportation. However, different cities also have similar needs that can be supported by IoT solutions. Table 14 presents an analysis of the growth of IoT applications in the different subcategories of the smart city according to consulting firms in the year 2018. They identify a significant growth in manufacturing with an average of 40%, followed by energy (23%), transportation (19.3%), and healthcare (10.6%) [90–93].

**Table 14.** Projections of IoT in smart city subcategories by 2025.

| Subcategory | Verizon | Growth Enabler | IoT Analytics |
| --- | --- | --- | --- |
| Manufacturing/Industry | 84% | 20% | 17% |
| Energy | 41% | 18% | 10% |
| Transportation | 40% | 7% | 11% |
| Healthcare | 11% | 20% | 6% |

### 5.4. Research Contribution

According to the qualitative analysis carried out in this work, Figure 19 shows the high level of interest of researchers in using the IoT technology in the field of smart energy and smart environment. Additionally, it is important to indicate that, even though there are important contributions related to smart waste management, there is still an important gap is this field; other situations offer opportunities to researchers.

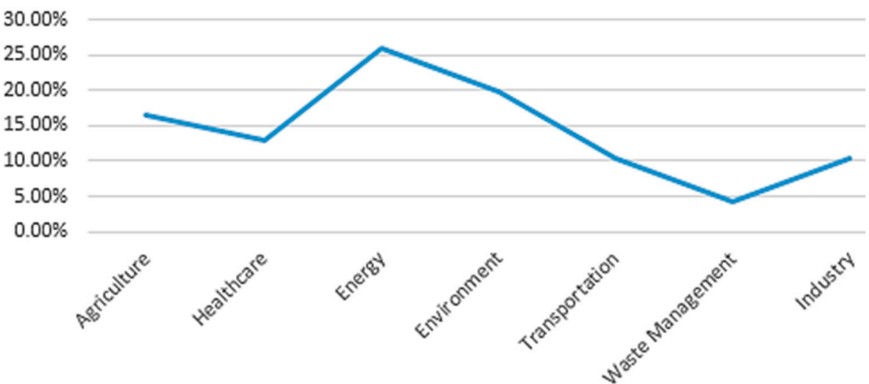

**Figure 19.** Rate of publications related to different smart cities subcategories.

Figure 20 shows that research works on smart energy has had a constant growth rate in the last three years. There is a significant increase in the rate of research works in smart environment and in smart transportation. The important growth of research works in the field of energy is associated with the increase of the number and variety of IoT smart devices (e.g., smart meters) and the automation of operations in power grids for developing complete smart grid architectures. These changes could be the basis for improving the economy of cities in the era of the fourth industrial revolution [94].

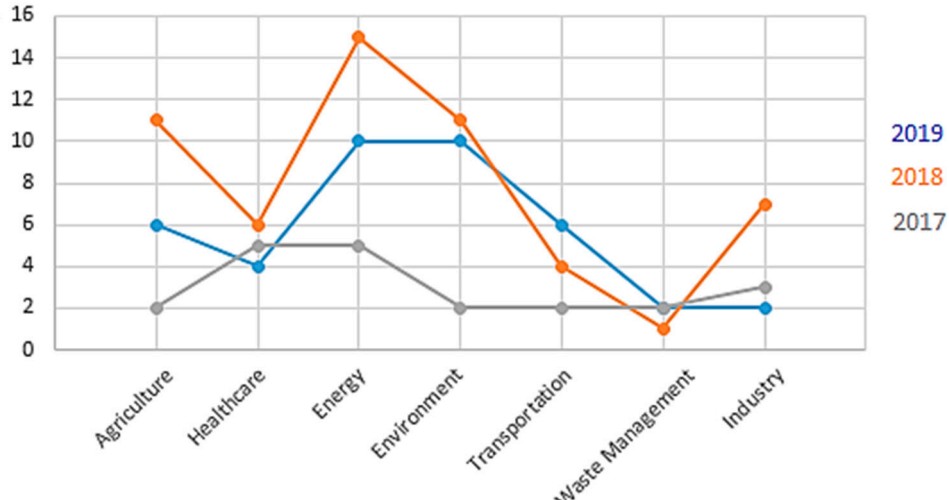

**Figure 20.** Rate of research works related to smart cities in the last three years.

The aim of the use of IoT in smart cities has been evolving. On the basis of the qualitative analysis carried out in this study, we have noticed that evolution has occurred at three levels.

Level 1, Monitoring: IoT solutions gather data from a specific part of the IoT ecosystem, e.g., temperature or humidity values of a farm.

Level 2, Automation: IoT solutions automate daily operations of the smart city, e.g., the control of an irrigation system.

Level 3, Intelligence for decision-making: IoT solutions deliver smartness to a city, e.g., the prediction of pest in plants based on data analysis.

Figure 21 shows the result of the word density analysis performed in the qualitative analysis. We identified that the most relevant aspect in the analyzed articles is the term "data". This is logic, since data are used at all levels (i.e., monitoring, automation, and intelligence) of IoT solutions.

**Figure 21.** Word density analysis in LoRa publications from 2015 to 2019.

According to our literature review, IoT research works started to include machine learning or data mining techniques to establish data analytic processes just a couple of years ago. Of the papers analyzed in this study, only around 35% of the publications considered the incorporation of data science approaches, and only 15% considered aspects related to decision support systems.

LoRa is an open, scalable, and flexible technology that can work with different protocols. These characteristics of LoRa make possible the implementation of the aforementioned three phases benefiting smart cities. Data generated for IoT solutions are sent using LoRa networks even to long distances. The data are sent in real time, allowing citizens and city managers to make effective decisions in a timely manner.

### 5.5. Challenges of LoRa in Smart Cities

According to our qualitative analysis, the inclusion of IoT solutions based on LoRa involves two relevant challenges for the industrial and research fields:

### 5.5.1. Security and Privacy

IoT devices gather data from critical infrastructures of the city, and such data are used in decision-making processes at the city administration level. Therefore, IoT solutions must ensure the privacy and security of the gathered data. However, the heterogeneity and limited capacity of devices make difficult the implementation of security systems. Although LoRa uses a security scheme for communication among different entities through of the use of security keys, most works considered in this study do not explain in detail the security aspects of their solutions. Only a few works indicate that they used the default security mechanism included in the LoRa standard.

In this sense, different topics related to IoT security such as threat and vulnerability detection, malware and intrusion detection, operating systems security, threat models/attack strategies in IoT ecosystem and IoT forensics are still a challenge in industrial and scientific fields.

IoT ecosystem should the use of security frameworks to assure and validate the security aspects. Authentication, confidentiality, and integrity should cover data, IoT devices, and external components such as fog, cloud, or web apps.

### 5.5.2. IoT Analytics

IoT systems generate large volumes of data which are processed using cloud or fog computing technologies. Data analytics can convert such data in valuable information that allow taking effective decisions. In other words, the synergy of IoT technology and data analytics allows the generation of real-time decision support systems.

It is important to indicate that some research proposals included data science approaches, and the most used data mining or machine learning techniques were decision tree (50% of cases), neuronal networks (ANN, 35% of cases), and time series (15% of cases).

We also identified a small minority of works including decision support systems (DSS); also we found that descriptive analysis was most commonly used; only a reduced number of works considered prescriptive analysis.

Taking into account that decisions at the city level have a strong impact on economic and social aspects, prescriptive analysis and DSS could be more adequate to get a complete visualization of smart city domains.

### 5.6. Strengths, Weaknesses, Opportunities, and Threats Analysis of LoRa in Smart City Applications

LoRa has been widely used in different smart city projects. The selection of LoRa in different cities was due to the strengths and opportunities that this technology offers. In Table 15, we present a Strengths, Weaknesses, Opportunities, and Threats (SWOT) analysis of LoRa based on the literature review performed in this work.

**Table 15.** SWOT analysis of LoRa in smart city applications.

| Strengths | Weakness |
|---|---|
| • Possibility of deploying private networks<br>• Low cost of hardware (less than $100)<br>• Low power consumption in non-transmission mode (less than 1 A)<br>• Long battery lifetime (more than 10 years).<br>• Large coverage (up to 15 km)<br>• More messages per day than other LPWAN technologies<br>• Robust to Doppler Effect and multipath fading, due to CSS modulation<br>• LoRa uses adaptive data rate (ADR)<br>• LoRa's sensitivity is on average 127 dBm<br>• Easy deployment and configuration. Many examples of configuration are available in Internet websites and scientific databases.<br>• End-to-end security is available in LoRa. | • Problems with line of sight in city due to buildings and trees.<br>• Many in-house or non-commercial solutions require low range coverage (less than 1 km).<br>• Applications with real-time interactions are not very suitable.<br>• Applications with data rate higher than 27 kbps are not suitable |
| **Opportunities** | **Threats** |
| • Deployment of LoRa networks in many cities around the world.<br>• New cities need a sensorization process.<br>• Big IT enterprises are developing or supporting LoRa products (e.g., Cisco, IBM, Intel).<br>• The need for smart city solutions using IoT is growing, especially in the healthcare, energy, industry, and transportation areas. | • Security attacks to IoT hardware and software<br>• Security issues due to lack of periodic updates of IoT devices<br>• Other strong IoT LPWAN technologies are available, like SIGFOX and NB-IoT |

*5.7. Impact of LoRa Research Proposals in the Smart City Initiative*

In this subsection, we analyze the contribution of research proposals related to IoT and LoRa to build a smart city in the context of a circular economy according to the New Urban Agenda approach. To carry out this analysis, three hypotheses were taken from reference [95]. These hypotheses were adapted for this study and are described below.

The first hypothesis is loose integration. in other words, the proposals of IoT–LoRa are focused on solving a specific need for a subcategory of the smart city, such as energy, water, health, but they have not considered a wider scope from the perspective of smart city. LoRa's proposals have been developed independently from the criteria or indicators of the smart city.

The second hypothesis is tight integration, in other words, IoT-LoRa proposals focus on covering a criterion or specific indicators of the circularity of an intelligent city. The proposals were developed in the context and following the guidelines of the smart city.

The third hypothesis is about data. The IoT-Lora proposals focus on the generation of data that can provide insights on certain criteria or indicators, based on which a decision can be made.

To perform the analysis, we classified 96 studies identified in the phase of inclusion of the PRISMA methodology (see the Table 16. This methodology was used in this study to produce a qualitative analysis based on the relationship of each study with one of the three above-mentioned hypotheses. We subsequently validated the dependence of each of the seven smart city subcategories defined in the same phase of inclusion on the hypotheses with a chi square independence test [96].

**Table 16.** Classification of papers by hypothesis.

| | Loose Integration | Tight Integration | Data | Total |
|---|---|---|---|---|
| Agriculture and Farming | 1 | 0 | 12 | 13 |
| Energy | 2 | 3 | 5 | 10 |
| Environment | 10 | 2 | 11 | 23 |
| Healthcare | 8 | 0 | 5 | 13 |
| Industry | 6 | 1 | 6 | 13 |
| Transportation | 3 | 3 | 5 | 11 |
| Waste Management | 3 | 3 | 7 | 13 |
| Total of paper | 33 | 12 | 51 | 96 |
| % Papers | 34% | 13% | 53% | |

Table 17 shows the contingence table used in the chi-square test. The following results were obtained: Chi 2 equaled 21.8, liberty grades (gl) equaled 12, and *p*-value was 0.0119. The hypothesis

to confirm is the independency of the application of the smart city from the grade of integration (loose, tight, data). According to the results of the independency test, the alternative hypothesis was selected. Thus, certain subcategories have a greater tendency to propose solutions related to smart city criteria and indicators, like the subcategory of energy. Other subcategories have a greater tendency to take advantage of the data generated to improve decision-making processes, like the subcategory of agriculture.

**Table 17.** Chi Square, between smart city subcategory, and three hypotheses.

|  | Loose Integration | Tight Integration | Data |
|---|---|---|---|
| Agriculture and Farming | 2.7 | 1.6 | 3.8 |
| Energy | 0.6 | 2.5 | 0.0 |
| Environment | 0.6 | 0.3 | 0.1 |
| Healthcare | 2.8 | 1.6 | 0.5 |
| Industry | 0.5 | 0.2 | 0.1 |
| Transportation | 0.2 | 1.9 | 0.1 |
| Waste Management | 0.5 | 1.2 | 0.0 |

These results led us to consider that, although the LoRa–IoT proposals focused on the environment subcategory developed between 2015 and 2019, the proposals were developed to cover a specific need in the field and were not used to enhance the strategies for building a smart city. Rethinking those approaches could contribute significantly, considering that the environment is one of the relevant axes within circular economy proposals. A second part of this result is that it could strengthen the use of data for decision-making processes in the subcategories of the smart city. This can generate new business models and research projects, especially in relation to the integration of IoT with data science.

Based on the results obtained from a qualitative analysis of the three hypotheses, it was possible to identify the relevant aspects of the proposals of tight integration and data that contribute to the development of a circular economy for the construction of a sustainable and smart city. To identify the relevant aspects, we used the concepts present in the works "Circular economy strategies in eight historic port cities: criteria and indicators towards an evaluation framework for circular cities" [2], and "The socio-economic integration of the circular economy are considered: An Integrative Framework" [97]. Table 18 presents the four relevant aspects obtained by this quality analysis, i.e., the social, economic, organizational, and material reuse aspects and the relation with the 96 studies considered in the inclusion phase of the PRISMA methodology of this study.

**Table 18.** Impact of IoT-LoRa proposals on circular economy of smart cities.

|  | Social | Economic | Organizational | Material Reuse |
|---|---|---|---|---|
| Agriculture and Farming | 3 | 1 | 7 | 2 |
| Energy | 1 | 4 | 2 | 3 |
| Environment | 16 | 2 | 4 | 1 |
| Healthcare | 7 | 0 | 4 | 2 |
| Industry | 0 | 7 | 3 | 3 |
| Transportation | 4 | 1 | 6 | 0 |
| Waste Management | 4 | 0 | 2 | 7 |

On the basis of the results obtained from (1) the analysis of the three hypotheses, (2) the contribution of the proposals to the four aspects of a circular economy and to the aspects identified in the subsections social, research, and industrial contributions, and (3) the SWOT analysis, we present the following possible academic and industrial impacts and future trends related to IoT–LoRa to build a smart city based on circular economy. Research focused on the use of LoRa and prompting the use of LoRa brings problems to the generation of a circular economy that adopts smart cities in order to obtain sustainability.

The qualitative analysis carried out of scientific proposals developed from 2015 to 2019 indicates that such works were designed to cover specific needs in the subcategories of agriculture and farming, energy, environment, healthcare, industry, traffic, waste management, which were defined in the "Materials and Methods" subsection of this study. The research proposals covered areas considered as critical (or key factors) in a circular economy, such as energy, mobility, environment, and communications, according to the "Circular economy strategies in eight historic port cities: criteria and indicators towards an evaluation framework for circular cities" [2] and "The socio-economic integration of the circular economy are considered: An Integrative Framework" [97].

The result of the analysis presented in the subsection called "Research Impact Contribution" show that most of the proposals delivered solutions for the energy, environmental, and transportation fields. In those works, the "communications" feature was just considered in an indirect way. For example, these works just considered basic features, such as the delivery of information gathered from the components of the IoT ecosystem to the user. LoRa-based proposals should also cover areas that are considered critical within the approach of a circular economy. For example, the works should consider the recycling or reuse aspects on which the circular economy is based to generate sustainability.

LoRa-based proposals make use of infrastructures or processes previously established in different fields such as health, agriculture, and energy. Based on this fact, the reuse of the elements is focused on incorporating IoT elements in these infrastructures to contribute with intelligence, as mentioned in the subsection "Research Impact Contribution". The objective of obtaining data is to generate intelligence for improving decision-making processes.

LoRa is an open standard. This feature allows IoT–LoRa solutions to adapt their technical and functional requirements according to smart city' needs, as mentioned in the subsection "Analysis of the Application Layer" of this work. LoRa devices can be re-used to improve functionalities for smart city applications by just adding some libraries or configurations. This adaptability or flexibility of LoRa identified in the "SWOT Analysis" subsection contributes significantly to the idea of circular economy in the smart city.

### 5.7.1. Future Trends of LoRa Proposals in Smart City Based on Circular Economy

On the basis of the analysis carried out using the methodology of the three hypotheses and the aspects of the circular economy, we identified some aspects that could be relevant for further studies and that could contribute significantly to the generation of a circular economy.

### 5.7.2. Data Integration Hypothesis

A relevant characteristic of previous works based on LoRa is data generation. It is covered by LoRa proposals in the four layers of the Y.2060 recommendation described in the "Materials and Methods" section of this study; however, the data are not used significantly in decision-making processes, as mentioned in the "Research Impact Contribution" section. Solutions seeking to complement their contribution with the transformation of data into intelligence for decision-making are shown in Figure 22.

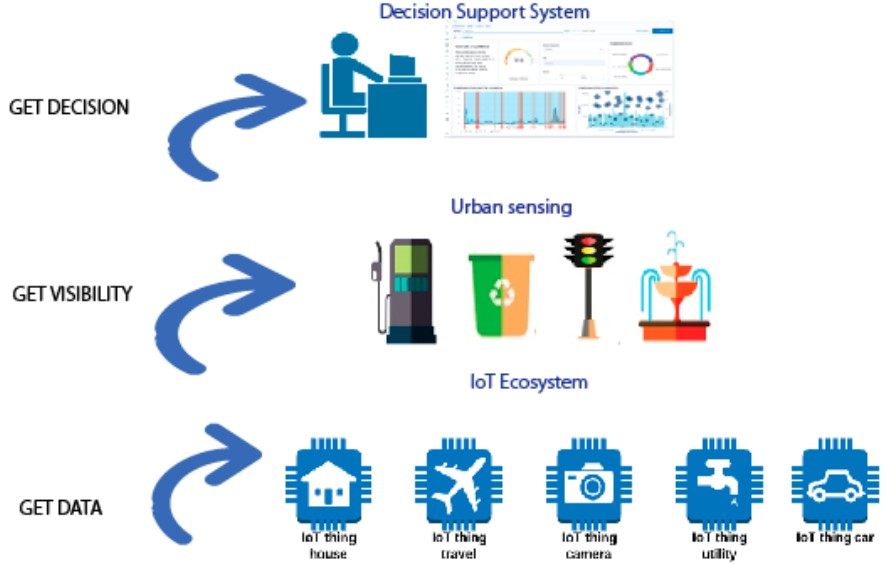

**Figure 22.** Three-phase cycle to generate intelligence for data decision.

### 5.7.3. LoRa Proposals for the Generation of New Business Opportunities

We classified the articles selected for this study as shown in Table 16, in which the contributions of LoRa's proposals are schematized from the perspectives of social, economic, organizational, and material reuse. We can say that the proposals delivered a great contribution for the improvement of particular organizational processes of each subcategory of the smart city. In the case of healthcare solutions, there was a greater focus on the social aspects, and most of the proposals did not consider the economic and material reuse aspects.

The circular economy in the smart city approaches the reuse of resources that are limited in nature, such as food or water. The proper management of these elements allows maintaining the sustainability approach of the city. The proposals related to energy management considered this feature, especially those focused on renewable energy and smart grids. Aspects such as the prediction of equipment maintenance or the degrading conditions of a product could contribute to this aspect of reuse, since they could allow the identification of the degrading conditions of a product and its substitution before its total degradation. This may be of interest to the manufacturing industry, because a completely non-damaged part can be reused to develop new components. Manufacturers can also start considering this aspect of reuse to design their products.

The circular economy drives new business models and interactions in the city. By using the data generated by IoT solutions, it is possible to generate direct benefits for the industry and for the local government. Manufacturers can, based on the user usage patterns of their products, find specific needs at a given time. They can consider in this way the creation of components that allow their reuse or the creation of new products from these. At the local government level, the determination of these patterns allows validating a service or understanding when it can be used by another citizen to optimize resources.

The sale of services is a new business model that has had a remarkable growth in recent years, which allows the reuse of existing infrastructures of companies and local governments significantly improving what is offered to the users or citizens. This is another context that could be strongly exploited by research and industry. Of the analyzed works, less than 1% focused on generating service sales models of LoRa IoT implementations.

### 5.7.4. LoRa Proposals in the Building of Smart Cities

A possible reason for not covering all of the indicators or criteria for the circular economy of smart city in the proposals presented from 2015 to 2019 may be the fact that they were not developed within the context of the smart city. According to the analysis carried out of the three hypotheses (loose integration, tight integration, and data), most contributions presented a loose integration. This indicates that although a specific need was being solved, it was not being considered within a macro-plan in the development of a sustainable city. A possible consideration in this regard would be to strengthen the interaction between the academy and local governments; also, research works should contribute to cover the proposed roadmaps for circular economy strategies for cities. Cities indicated in the subsection "Smart City Strategies Using IoT LoRa" have established a roadmap for circular economy by the year 2021.

### 5.7.5. Aspects to Consider to Develop IoT Solutions that Contribute to the Development of Smart City Circularity.

Tight or data integration in IoT-LoRa proposals can contribute to the circular economy in smart cities. Some considerations from the study are presented below.

1. The criteria or indicators of circular economy or circularity related to the smart city sector for which the solution is developed should be considered.

2. Identify the infrastructure of the city related to the sector that could use the developed solution.

3. Analyze the technological, legal, or economic barriers that may affect the development of the solution.

4. Identify the infrastructure of the city to which it is possible to add the intelligence component for improving decision-making processes.

5. Design the solution under the scheme of re-using, according to the concepts of circular economy, which will allow adding functionalities to the smart city solution without having to invest in new components.

6. Configure the technical aspects of the components of the IoT ecosystem (devices, LoRa, fog, or cloud) based on the functional characteristics of the smart city subcategory. Select the most appropriate aspects of SF, data rate, and throughput for each case.

7. Validate the security requirements that guarantee the confidentiality, integrity, and availability of the implemented solutions.

## 6. Conclusions

The qualitative analysis carried out of scientific IoT–LoRa proposals from 2015 to 2019 indicates that these works were aligned to cover specific needs in the subcategories of agriculture, energy, environment, healthcare, industry, traffic, and waste management. However, according to the hypotheses presented in this work, most of the IoT solutions based on LoRa were integrated in a loose way. Thus, although a specific need of a subcategory of the smart city was being solved, it was not considered within a macro-plan for the development of a sustainable city. Rethinking the approaches of LoRa–IoT proposal on the basis of circular economy could contribute significantly to a macro-plan for the development of a sustainable city. The circular economy drives new business models and interactions in the city. By using the data generated by IoT solutions, it is possible to generate direct benefits for the industry and for local governments.

In this paper, a qualitative analysis of LoRa was carried out about the development of smart city solutions related to different aspects of the city such as transportation, health monitoring, and pollution level measurements. For the qualitative analysis process, a systematic literature review of papers found in five scientific databases was executed, following the PRISMA methodology. Then, we analyzed the relevant aspects of LoRa guided by the IoT architecture model proposed by the ITU in its Y.2060 recommendation, in order to establish the strengths, weaknesses, opportunities, and threats (SWOT)

that support the selection of LoRa as the technology for the connectivity of IoT devices in smart city models.

On the basis of this literature review, projects related to smart cities and LoRa were identified, which were classified in the smart health, smart traffic, smart environment, smart agriculture, and smart energy subcategories. Subsequently, we analyzed the projections of consulting firms in relation to the deployment of IoT for the next years till 2025, finding that the type of applications that are growing are similar to the subcategories that were identified in the literature review of this work, which allows us to assume that the research projects in the academic field are aligned with the current social and business needs. This coincides with the recent increase of the number of papers.

LoRa is an unlicensed technology that allows creating a private network, and this is one of the main advantages of this technology. It also has other advantages such as a strong modulation scheme that makes it resistant to the Doppler effect, multipath fading, scalability in terms of number of nodes, adaptability to data frame length, and an end-to-end security architecture. These strengths allow LoRa to be one of the most selected LPWAN's technologies for the development of smart cities around the world. However, some security aspects, such as response to security attacks and vulnerability policies, still need be solved.

The development of LoRa solutions present an industrial impact related to the projections in different subcategories of smart city such as energy, healthcare, and transportation; according to the predictions of recognized consulting firms in the world, increments of approximately 10% per year until 2025 are present in each subcategory of smart city, and higher investments are projected in the following years. In the academic field, there is a significant increase in the development of IoT LoRa solutions, which is reflected by the increased number of scientific publications, from about 30 scientific articles in 2016 to about of 400 in 2018.

LoRa is an open, scalable, and flexible technology that allows interoperability between five relevant components for the construction of an intelligent city that considers sensors, complementary technologies, data science, citizen interaction, LoRaWAN, in order to obtain data and generate visibility on aspects related to subcategories of the smart city, such as smart health, smart agriculture, smart transportation, among others. The visibility generated seeks to provide support for decision-making in an effective and timely manner.

The inclusion of LoRa seeks to solve five challenges in smart cities: productivity improvement, predictive maintenance, energy management, monitoring of human health, and operational efficiency.

On the basis of our qualitative analysis, the inclusion of LoRa poses two relevant challenges:

1. Security and privacy in IoT environments that protect the generated data and critical infrastructure.

2. The inclusion of data analytics in IoT environments to take advantage of the generation of data to improve decision-making processes.

**Author Contributions:** Conceptualization, R.O.A., and S.G.Y.; methodology, R.O.A., and S.G.Y.; formal analysis, R.O.A., and S.G.Y.; investigation, R.O.A., and S.G.Y.; writing–review and editing, R.O.A., and S.G.Y.; project administration, S.G.Y.

**Funding:** This research was funded by Escuela Politécnica Nacional through the project PIJ-17-08-"Diseño e Implementación de un Sistema de Parqueadero Inteligente".

**Acknowledgments:** The authors gratefully acknowledge the financial support provided by the Escuela Politécnica Nacional, for the development of the project PIJ-17-08-"Diseño e Implementación de un Sistema de Parqueadero Inteligente".

**Conflicts of Interest:** The authors declare no conflicts of interest.

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
