# Peer review of "A Comprehensive Study of the Use of LoRa in the Development of Smart Cities"

_applsci, doi:10.3390/app9224753_

Round 1
Reviewer 1 Report
A positive aspect of the contribution is that the topic is of interest and actuality. The paper is well documented and written regarding the LoRa in general and introductory concepts.
Many of the considerations given in the chapter 3. “Result of Analysis of Works” are generally for IoT applications such as agriculture, healthcare, energy, industry, safety, environment and traffic and are presented only superficially without technical details and implementable solution.
A significant part of the paper describes different concepts in a manner similar to a lecture:
- Materials and Methods; Stage 1: Identification; Stage 2: Screening, Stage (2 or 3): Eligibility Analysis, Stage 4: Inclusion;
- Result of Analysis of Works; Analysis of the Application Layer (Agriculture, HealthCare, Traffic control and Transportation, Energy, Environment, Waste management); Analysis of the Network and Transport capabilities (Data Layer, Coverage of LoRa in IoT Applications); Analysis of the Device Layer (Sensors, Nodes).
The work is more likely addressed to beginners in the field of LoRa, IoT, not only to experts and this is not an advantage.
Why are DHT11 and DHT22 sensors important for LoRA?
Why is important ESP8266 Wi-Fi module for LoRA? ESP8266 is an embedded system with Wi-Fi, while SX1272 or SX1276 are transceivers for LoRA.
Why are presented relations DR, Ttx, …, Noise Floor?
What do you mean with: “Low power consumption in node devices; in non-transmission mode (less than 1 A)”?
The paper is well suited for a book chapter or magazine article, showing LoRa in general and IoT introductory concepts.
Author Response
Dear reviewer
Thank you for your suggestions. We send in attachment the content added to improved the revised manuscript.

Reviewer 2 Report
The authors have presented a review on the use of LoRa in the development of smart cities. The study is very interesting and covers different aspects and subcategories of smart cities. The paper is well structured, easy to read, and the references are recent.
I believe that this paper contributes to the understanding of the application of LoRa in smart cities.
I have some minor comments which are listed in the following.
The background section could be divided into two subsections There are some redundant materials. For example, the sentences from line 153 to 159 could be omitted or simplified, same for lines 240 to 249. Make sure all abbreviations are defined in the text. It could be interesting to see the journals or the top 10 journals related to figure 8. Figure 10 is interesting but it could be more interesting to have the number of papers by smart city application domains, or to mention the references on each application domain of the figure 10. Figures 11 to 16 are not clear, for example, for figure 11, it is mentioned Components of IoT architecture for agriculture, but there are both technical components and agriculture subcategories. What is the link between them? In the discussion section (section 4) some smart cities are presented in figure 21, but only the applications of Manchester and Montevideo are described, what about the others? It could be interesting to provide a small description about the applications of smart city in the cities mentioned in figure 21 (maybe a table). Some references are needed in the paper: figure 1, line 80 to 90, line 98 to 101, figure 5, line 140 to 145, figure 17?, line 434 to 443, line 537 to 538, line 618 to 621, line 609 to 616, figure 21, line 648 to 652Author Response
Dear reviewer
Thank you for your suggestions. We send in attachment the content added to improved the revised manuscript.

Reviewer 3 Report
This study presents a comprehensive review of the utilization of LoRa technology in smart cities worldwide. As one of the most popular LPWAN technologies, the LoRa technology has strong capability in connecting multiple IoT nodes with low power consumption and long communication range. The primary contribution of this study is that it provides a comprehensive understanding of the applications, compliant hardware and services of the LoRa technology. The topic of this study is significant to the users of the LoRa technology and decision makers of smart cities.
This manuscript is organized in a clear and easy-understanding way. Background section provides critical concepts of smart cities and characteristics of the LoRa. Methods used in the study are reasonable to address the problem. Results and discussion can reflect the objectives of the study. However, results and discussion section needs to be improved, and academic contributions of the study should be highlighted in the Abstract and conclusions section.
Major Comments
(1) Can you merge Figures 10 -16 to a figure? Information between Figure 10 and other figures is duplicated. In addition, in the merged figure, I recommend organizing the components in a more logical way, such as using a hierarchical figure. For instance, there are ten components of agriculture sector, and they can be reclassified in to a few categories. Why some components are linked and others are not should be explained in the manuscript.
(2) Can you reorganize the discussion section? It is difficult to find the logic of discussion section. Please reorganize it with a few subsections.
(3) What is the major academic contributions of this study? This study provides a comprehensive understanding of the utilization of LoRa, but the summary is not academic contributions. The academic contributions should be innovative analysis and future recommendations. In addition, what is the industrial impacts of the study? Following papers may be referred to answer the above two questions. “Trends and opportunities of BIM-GIS integration in the architecture, engineering and construction industry”, “Future effectual role of energy delivery: A comprehensive review of Internet of Things and smart grid”, and “Low-power wide area network technologies for Internet-of-things”. Finally, please discuss the academic contributions, future recommendations, impacts of the study on industries, significance and innovations in the discussion section. Then, summarize and highlight them in the Abstract and conclusion sections.
Author Response

(The authors gave the same response as above.)

Round 2
Reviewer 1 Report
I followed the review questions and how the authors responded and made the necessary improvements to the paper content. The authors have clarified all the aspects and they have modified the paper accordingly.
Author Response
Dear Review:
We are very grateful for your comments for our manuscript. Thanks for all your suggestions.
Reviewer 3 Report
In the current version, parts of the manuscript have been revised in terms of the comments and suggestion, but extensive revisions are still required for the two comments in the last review. Minor revisions to the two comments are not enough.
(1) Figures 10 - 16 have been merged as Figures 10 and 11, but hierarchies in the figures and more explanations are still required in the manuscript. For example, terms "waste", "transport", "agriculture", etc. should be placed on a higher hierarchy than terms "methane", "load", etc. Different colors are recommended to present terms on various hierarchies. In addition, terms listed on figures should be explained in the manuscript.
(2) A Research Impact section is added in the revised manuscript, but the major academic contributions of the study are not included in the section. The articles you referred in the revised manuscript are not listed as references. In addition, as I mentioned in the comments in last review, please discuss the academic contributions, future recommendations, impacts of the study on industries, significance and innovations in the discussion section. You need to discuss the contributions and innovations based on your studies instead of summarizing other researches again. Then, summarize and highlight them in the Abstract and conclusion sections.
Author Response
Dear Review
We are very grateful for your comments for our manuscript. We have revised the manuscript in accordance with your comments and send you in attachment.

Round 3
Reviewer 3 Report
Thanks for your revision. Authors have added paragraphs of statements to explain their responses to the comments. However, the added statements are not closely related to the comments and suggestion. Can you kindly carefully read the previous comments and suggestion included in the last two review reports? Again, I believe the topic of this study is critical to the technical development and applications of smart cities, but the quality of the study has to be significantly improved before publication. Therefore, I strongly suggest authors carefully response each comment. Responses are not adding too many statements, but to summarize and highlight contents that are not mentioned in previous versions.
Author Response
Dear reviewer 3
We are very grateful for your comments for our manuscript, in attached you will find our response to your comments.
Thanks again for your valuable suggestions.

Round 4
Reviewer 3 Report
Thanks for your careful revision. The issues mentioned in last three versions of review reports have been addressed. In addition, academic contributions, significance and innovations of the study have been highlighted and reorganized in the current version. From my perspective, contents of the current version can be accepted. English language needs to be edited.
Minor comments:
Line 64: Grammar error needs to be revised.
Lines 20 and 66: Can you check if "open characteristic" is a correct presentation here?
Line 1017: The sentence should be simplified.
Line 1034: Grammar error of this sentence should be revised.
Line 1133: Grammar error of the sentence "However, ..." should be revised.
Author Response
Dear Reviewer
We are very grateful for your comments for our manuscript. We have revised the manuscript in accordance with your comments. We send our response in attachment. Thanks for your suggestions.
